



# Impacts of atmospheric transport and biomass burning on the interannual variation in black carbon aerosols over the Tibetan Plateau

Han Han[1,#], Yue Wu[1,2,#], Jane Liu[3,1], Tianliang Zhao[4], Bingliang Zhuang[1], Yichen Li[1], Huimin Chen[1], Ye Zhu[5], Hongnian Liu[1], Qin'geng Wang[6], Shu Li[1], Tijian Wang[1], Min Xie[1], and Mengmeng Li[1]

[1]School of Atmospheric Sciences, Nanjing University, Nanjing, China

[2]Suzhou Meteorological Bureau, Suzhou, China

[3]Department of Geography and Planning, University of Toronto, Toronto, Canada

[4]School of Atmospheric Physics, Nanjing University of Information Science & Technology, Nanjing, China

[5]Shanghai Public Meteorological Service Centre, Shanghai, China

[6]School of the Environment, Nanjing University, Nanjing, China

[#]These authors contributed equally to this work.

Correspondence: Jane Liu (janejj.liu@utoronto.ca)

## Abstract

Atmospheric black carbon (BC) in the Tibetan Plateau (TP) can largely impact regional and global climate. Still, studies on the interannual variation in atmospheric BC over the TP and associated variation in BC sources and controlling factors are rather limited. In this study, we characterize the variations in atmospheric BC over the TP surface layer through analysis of 20-year (1995-2014) simulations from a global chemical transport model, GEOS-Chem. The results show that, of all areas in



the TP, surface BC concentrations are highest over the eastern and southern TP, where surface BC are

susceptible respectively to BC transport from East Asia and South Asia. Combining the GEOS-Chem

simulations and trajectories from the Hybrid Single-Particle Lagrangian Integrated Trajectory

(HYSPLIT) model, we assess the contributions of different source regions to surface BC in the TP. On

the 20-year average, over 90% surface BC in the TP comes from South Asia (47%) and East Asia (46%).

Regarding seasonal variation in foreign influences, South Asia and East Asia are dominant source

regions in winter and summer, respectively, in terms of both magnitude and affected areas in the TP. In

spring and autumn, the influences from the two source regions are somewhat comparable. Interannually,

surface BC over the TP is largely modulated by atmospheric transport of BC from foreign regions year-

round and by biomass burning in South Asia, mostly in spring. We find that the extremely strong

biomass burning in South Asia in the spring of 1999 greatly enhanced surface BC concentrations in the

TP (31% relative to the climatology). We find that the strength of the Asian monsoon correlates

significantly with the interannual variation in the amount of BC transported to the TP from foreign

regions. In summer, strong East Asian summer monsoon and South Asian summer monsoon tend to,

respectively, increase BC transport from central China and northeast South Asia to the TP. In winter, BC

transport from central China is enhanced in years with strong East Asia winter monsoon or Siberian

High. A strong Siberian High can also increase BC transport from northern South Asia to the TP. This

study underscores the impacts of atmospheric transport and biomass burning on the interannual

variation in surface BC over the TP. It reveals a close connection between the atmospheric transport of

BC from foreign regions to the TP and the Asian monsoon.

## 1 Introduction

Black carbon (BC) is a carbonaceous aerosol formed from combustion of carbon-based fuels and

materials. BC in the atmosphere is a major air pollutant and a strong absorber of solar radiation (Bond et



al., 2013). Atmospheric BC can greatly influence regional (Ramanathan and Carmichael, 2008; Zhuang
et al., 2018) and global (Allen et al., 2012; Chung et al., 2012) climate through multiple mechanisms. It

can cause atmospheric heating (Cappa et al., 2012) and surface dimming (Flanner et al., 2009) and
influence cloud formation and development processes (Jacobson et al., 2012). Furthermore, after its
deposition on snow or ice, BC reduces the surface albedo and accelerates the melting of glaciers and
snow cover (Hansen and Nazarenko, 2004; Flanner et al., 2007).

The Tibetan Plateau (TP) has an average altitude of over 4 km and an area of 2.6 million km$^2$,
known as the Third Pole. Because of its special geography, the TP can greatly impact regional and
global climate through dynamic and thermal processes (Wu et al., 2015; Li et al., 2018). The TP has a
large number of glaciers and a wide coverage of snow. Although atmospheric BC in the TP is among the
lowest in the world, BC there can alter the climate (Lau et al., 2010; Jiang et al., 2017), ecosystem

(Kang et al., 2019), and hydrology (Barnett et al., 2005) in the TP, consequently influencing the living
environment of billions of people in the world. Atmospheric BC is an important factor driving the
surface warming in the TP due to its strong absorption of solar radiation (He et al., 2014a). After its
deposition to the TP ground, BC in the snow reduces the surface albedo (Ming et al., 2009; Qian et al.,
2011; Qu et al., 2014) and the duration of snow cover (Ménégoz et al., 2014; Zhang et al., 2018) in the

TP. Furthermore, BC in both atmosphere and cryosphere over the TP is responsible for retreats of the
snow cover (Menon et al., 2010; Xu et al., 2016) and glaciers (Xu et al., 2009; Ming et al., 2012; Niu et
al., 2020).

        Atmospheric BC concentrations in the TP vary with location and season, which was revealed by

limited observations over different regions in southern (Marinoni et al., 2010; Putero et al., 2014; Chen
et al., 2018), northern (Zhao et al., 2012), and southeastern (Cao et al., 2011; Wang et al., 2018, 2019)



TP. Over these regions, seasonal variations in atmospheric BC show different patterns. In the Himalayas over the southern TP, surface BC concentrations were observed to reach the highest in spring and the lowest in summer (Marinoni et al., 2010; Chen et al., 2018). In the Qilian Shan over the northern TP, surface BC concentrations were reported to be highest in summer and lowest in autumn (Zhao et al., 2012). Wang et al. (2016) suggested that surface BC concentrations show a seasonality of winter high and spring low at a site in the southeastern TP, while a pattern of winter low and spring high at a site in the central TP. Few studies have explored the interannual variation in atmospheric BC concentrations over the TP (Mao and Liao, 2016). Since BC observations in the TP are limited due to the harsh environment and sparse sites, numerical modeling on the variation in atmospheric BC over the TP is desirable.

Due to weak anthropogenic activities and biomass burning, the contribution of local emissions to atmospheric BC in the TP is low (Zhang et al., 2015). Concentrations of atmospheric BC in the TP are greatly influenced by the long-range transport of BC from foreign regions (Kopacz et al., 2011; Kang et al., 2019). Previous studies have investigated the pathways of BC transport to the TP (Cao et al., 2011) and some of them suggested that South Asia and East Asia are two main source regions of atmospheric BC in the TP (Lu et al., 2012). The Asian summer monsoon system was identified as an important influencing factor for BC transport from South Asia to the TP (Chen et al., 2013; Han et al., 2014; Xu et al., 2014; Zhang et al., 2015). In summer, BC from northern India can be transported to the middle and upper troposphere and then crossing the Himalayas to the TP via southwesterly winds (Yang et al., 2018). BC emissions in East Asia can also be uplifted to upper layers by summer monsoon circulation and then transported to the northeastern TP (Zhang et al., 2015). The midlatitude westerlies are favorable for BC transport from central Asia and northern India to the western TP (Chen et al., 2018) but unfavorable for BC transport from eastern China to the TP (Cao et al., 2011). Although previous





studies explored the mechanisms of BC transport to the TP, large uncertainties remain in the quantified fractional contributions of BC transport from different source regions to the TP (Yang et al., 2018). More importantly, how BC transport to the TP varies interannually and what are underlying mechanisms for the variation are unclear. Therefore, it is necessary to examine how seasonal BC transport to the TP

varies from year to year and whether there is a connection between the Asian monsoon and the interannual variation in BC transport to the TP.

Observations and simulations showed previously that anthropogenic and fire emissions are major sources of atmospheric BC in the TP (Lu et al., 2012; Zhang et al., 2015). Zhang et al. (2015) estimated

that biomass burning together with biofuel emissions can contribute to around half of the annual mean BC column burden over the TP. Engling et al. (2011) reported that BC emissions from fire events in Southeast Asia in spring could probably increase the BC concentrations over a mountain site in the southeastern part of the TP. Putero et al. (2014) suggested that over half of the high BC episodes in the southern Himalayas were likely affected by the fire events in South Asia. However, these studies

demonstrated the influences of biomass burning in a relative short term or during some fire events, few investigated the influences in a long term over a decade. The influence of biomass burning on the interannual variation in atmospheric BC over the TP warrants an in-depth study.

In this study, we aim to assess the impacts of atmospheric transport and biomass burning on

surface BC concentrations over the TP, especially on the interannual variation in BC during 1995-2014. To estimate BC transport from different source regions to the TP, we adopt a numerical approach based on a global chemical transport model, GEOS-Chem (Bey et al., 2001), and a trajectory model, the Hybrid Single-Particle Lagrangian Integrated Trajectory model (HYSPLIT) (Draxler and Hess, 1998; Stein et al., 2015). In the following, the method and models are described in section 2. Section 3





discusses the seasonal variations in surface BC over the TP and in BC transport from source regions to

the TP based on the mean status of the 20-year simulations. The interannual variation in surface BC

over the TP and the impacts of biomass burning and transport on this variation are analyzed in section 4.

Conclusions, along with discussion, are provided in section 5. In this paper, BC refers to BC aerosols in

the atmosphere. Surface BC refers to atmospheric BC aerosols in the surface layer.


## 2 Data and methods

### 2.1 GEOS-Chem simulations

A global chemical transport model, GEOS-Chem (version v9-02, http://geos-chem.org) (Bey et al.,

2001), is used in this study to simulate global BC concentrations. GEOS-Chem is driven by the NASA

Modern-Era Retrospective Analysis for Research and Applications (MERRA) meteorological data

(Rienecker et al., 2011). In this study, we focused on how surface BC in the TP responds to interannual

variations in natural processes including biomass burning and meteorology. Therefore, anthropogenic

emissions in our simulation were allowed to vary seasonally but not interannually, i.e. anthropogenic

emissions in 2000 including their seasonality were used for 20 simulation years. We conducted three

GEOS-Chem simulations in this study: CTRL, FixBB, and FixMet. The three simulations covered the

period from 1995 to 2014 (using 1994 for spin-up) at $2^o$ latitude by $2.5^o$ longitude horizontal resolution

with 47 vertical layers. In CTRL, both biomass burning emissions and meteorological fields varied

interannually. In FixBB, interannual meteorology was allowed and fire emissions were fixed in 2005, so

to remove the impact of the interannual variation in biomass burning. In FixMet, emissions from

biomass burning were allowed to vary interannually and meteorology was fixed in 2005, so to remove

the impact of interannual meteorology.

      In the simulations, global anthropogenic BC emissions were based on Bond et al. (2007), with an



annual emission of 4.4 Tg C in 2000 (Leibensperger et al., 2012). Global biomass burning emissions of

BC were from the Global Fire Emissions Database version 3 (GFED3) inventory (van der Werf et al.,

2010), which covers the period of 1997-2011. BC in GEOS-Chem is represented by two tracers:

hydrophobic and hydrophilic (Park et al., 2003). Freshly emitted BC is mostly (80%) hydrophobic

(Cooke et al., 1999). Hydrophobic BC becomes hydrophilic typically in a few days (McMeeking et al.,

2011), which is simply assumed as 1.15 days in the model, called an e-folding time (Cooke et al., 1999;

Park et al., 2005; He et al., 2014b). Simulations of aerosol dry and wet depositions follows Liu et al.

(2001). Dry deposition of aerosols is simulated using a resistance-in-series model (Walcek et al., 1986)

dependent on local surface type and meteorological conditions, while wet deposition scheme includes

scavenging in convective updrafts, as well as in-cloud and below-cloud scavenging from convective and

large-scale precipitation. Dry deposition is generally smaller than wet deposition (He et al., 2014b; Li et

al., 2015). Tracer advection is computed every 15 minutes with a flux-form semi-Lagrangian method

(Lin and Rood, 1996). The tracer moist convection scheme follows Allen et al. (1996a, b), using GEOS

convection, entrainment, and detrainment mass fluxes. The deep convection is parameterized using the

relaxed Arakawa-Schubert scheme (Arakawa and Schubert, 1974; Moorthi and Suarez, 1992) and for

the shallow convection, the scheme in Hack (1994) is used.


**2.2 GEOS-Chem evaluation**

GEOS-Chem simulations of surface BC concentrations were previously evaluated over the TP and

China (He et al., 2014b; Li et al., 2015). Here, we validated the model in the TP and its surrounding

regions for enhanced confidence. As surface BC measurements in the TP are rather limited, the

observation data collected at 13 sites (Figure 1) from previous literature (Carrico et al., 2003; Qu et al.,

2008; Zhang et al., 2008; Beegum et al., 2009; Ganguly et al., 2009; Bonasoni et al., 2010; Ming et al.,

2010; Pathak et al., 2010; Ram et al., 2010a, b; Nair et al., 2012) were used in this study, following He



et al. (2014b). The 13 sites were grouped into urban, rural, and remote sites (He et al., 2014b). The

observational data are available for 2006 at 9 of the 13 sites and available at the other sites for different

periods, i.e., 1999-2000, 2004-2005 and 2008-2009. BC observations at a remote site during 2015-2017

from another study (Chen et al., 2018) were also used.

The annual mean surface BC concentrations from GEOS-Chem and observations are compared in

Table 1. The observed surface BC concentrations are below 2 μg m$^{-3}$ at remote sites, about 2-5 μg m$^{-3}$ at

rural sites, and as high as 5 μg m$^{-3}$ at urban sites (He et al., 2014b). Compared with the observations,

GEOS-Chem performs well at the remote sites, moderately at the rural sites, and poorly at the urban

sites (Table 1). The simulations substantially underestimate surface BC concentrations at the urban sites,

likely due to the coarse horizontal resolution in the model that dilutes the intensity of local emissions in

a model grid. Taking the rural and remote sites only (Figure 2a), we found a high consistency between

the annual mean simulations and observations, with a significant correlation coefficient of 0.99. The

comparison suggests that GEOS-Chem can generally capture the spatial variation in surface BC

concentrations over the TP. Moreover, the seasonality of simulated surface BC concentrations was

evaluated at three sites (Figures 2b-2d). GEOS-Chem simulates low BC concentrations in summer and

high BC concentrations in winter and spring at the sites. The amplitude of the seasonal variation in the

simulations is weaker than that in the observations. Moorthy et al. (2013) found that simulated surface

BC concentrations by the Goddard Global Ozone Chemistry Aerosol Radiation and Transport

(GOCART) model in winter were lower than the observed ones at a TP site and they attributed this to

the biases in the atmospheric boundary layer parameterization scheme. Wintertime surface BC

concentrations were also underestimated by the Community Atmosphere Model version 5 (CAM5,

Zhang et al., 2015), suggesting a common bias in these models.



### 2.3 Meteorological and fire data

The meteorological data used in this study are the NCEP/NCAR (National Centers for Environmental Prediction/National Center for Atmospheric Research) reanalysis, available from the Physical Sciences Division of NOAA Earth System Research Laboratory (https://www.esrl.noaa.gov/psd/data/gridded/data.ncep.reanalysis.surface.html). The data include geopotential height and wind. The horizontal resolution is 2°latitude $\times 2.5^{\circ}$ longitude.

The nighttime fire count product retrieved from ATSR (Along Track Scanning Radiometer) using Algorithm 2, available from European Space Agency (http://due.esrin.esa.int/page_wfa.php), were used to verify the biomass burning emissions in the model. ATSR is onboard the Second European Remote-Sensing Satellite (ERS-2). The spatial resolution of the data is 1 km, and the sensor achieves a global coverage every three days. The ATSR satellite data with the period of 1997-2011 were gridded to the GFED3 grids with a resolution of $0.5^{\circ} \times 0.5^{\circ}$ in longitude and latitude.

### 2.4 Transport estimation

Combining GEOS-Chem simulations and HYSPLIT (version 4, http://www.arl.noaa.gov/HYSPLIT_info.php, Draxler and Hess, 1998; Stein et al., 2015) trajectories, we estimated the contributions of different source regions in the world to surface BC in the TP during 1995-2014. HYSPLIT is an atmospheric transport and dispersion model (Fleming et al., 2012), developed by the Air Resources Laboratory of the National Oceanic and Atmospheric Administration (NOAA). Meteorological inputs to HYSPLIT are the NCEP/NCAR reanalysis at a resolution of $2.5^{\circ}$ latitude $\times 2.5^{\circ}$ longitude. We evenly divided the TP into 70 GEOS-Chem grids. Considering that the average lifetime of atmospheric BC is about a week, we simulated 7-day backward trajectories originated from each of the 70 grids. The trajectories were initialized four times a day (00, 06, 12 and 18





UTC) during 1995-2014. The starting altitude for the trajectories is 100 m above ground which is within

the typical planetary boundary layer in the TP (Ram et al., 2010b). We divided the world into six regions

(Figure 1b), including central Asia, East Asia, South Asia, Southeast Asia, the region of other Asia,

Europe, and Africa, and the rest of the world. BC concentrations from CTRL simulation were used in

the estimation.

Lu et al. (2012) proposed a novel approach that combined BC emissions with backward trajectories

to quantify the origins of BC in the TP. Modifying Lu et al. (2012)'s approach, we combined BC

concentrations, instead of BC emissions in Lu et al. (2012), with backward trajectories for the same

purpose. We assume that BC aerosols have a lifetime of $D$ days and the back trajectories are simulated

for $D$ days ($D=7$ in this study). To make the estimation stable, the amount of BC transported to a TP

surface grid on a day is assumed to be a mean of the BC transport along the backward trajectories

originated from that grid in the past $D$ days, i.e.,

$$BC_{imported} = \frac{\sum_{d=1}^{D} BC_d}{D} \qquad (1)$$

where $BC_d$ is the amount of BC that are transported to that TP surface grid along the backward

trajectory on a previous day $d$ $(d=1, 2, ... D)$.

Equation (1) provides a way to estimate the amount of BC that is transported to the TP from any

model grid outside the TP during a period of interest. For a grid $g_{i,j,k}$, the total amount ($C_{i,j,k}$) of BC

transported from $g_{i,j,k}$ to the TP is estimated by

$$C_{i,j,k} = \frac{\sum_{n=1}^{N} c \times v}{D \times M} \qquad (2)$$

where $i$, $j$, $k$ are indices for the model grid in longitude, latitude, and altitude coordinates, respectively. $n$

is an index for the number of trajectories. $N$ is the total number of trajectories that have passed through





the grid $g_{i,j,k}$ during the period of interest, for example, in a month. $c$ is the daily BC concentrations at

$g_{i,j,k}$ when trajectory $n$ passing $g_{i,j,k}$, and $v$ is the volume of $g_{i,j,k}$. $M$ is the number of trajectories in a day

($M$=4 in this study). Therefore, the total amount of BC transported to the TP ($T_{i,j}$) from the entire

tropospheric column above a surface grid $g_{i,j,0}$ in a source region during the period of interest is assessed

by

$$T_{i,j} = \sum_{k=1}^{K} C_{i,j,k} \quad (3)$$

where $K$ is the number of model layers in the troposphere.

Finally, the amount of BC transported from a source region to the TP surface can be summed up

and the fractional contributions of different source regions to surface BC in the TP can be quantified.

This method is inspired by Lu et al. (2012) and is robust and stable because it is not sensitive to the

number of trajectories taken in a day ($M$) and the number of days taken for the trajectories ($D$).

Using this method, the amount of BC transported from a source region to the TP surface is

determined by both BC concentrations over that region and the number of trajectories passing through

that region within the tropospheric column. An example of the estimation of BC transport to the TP

surface in April 2005 is shown in Figure 3. In this example, BC concentrations are high in central China

around 110°E (Figure 3a). A large number of trajectories pass through central China and finally arrive at

the TP surface (Figure 3b). Therefore, the amount of BC transported from central China to the TP is

high (Figure 3c). In contrast, although trajectories pass through central Asia are also in large numbers,

the amount of BC transported from central Asia to the TP is small (Figure 3c) because of the low BC

concentrations over central Asia and the trajectories appear at high altitudes (Figures 3a and 3b).



## 3 Seasonal variations in surface black carbon over the Tibetan Plateau and in black carbon transport to the Tibetan Plateau

BC transport from each of the source regions to the TP surface varies with season. Figure 4 shows the amount of BC transported from each GEOS-Chem grid to the TP surface in the four seasons. Obviously, surface BC in the TP mainly originates from South Asia and East Asia, especially from the regions near the southern and eastern borders of the TP, including central China, northeastern South Asia, and northern South Asia. These spatial distributions of BC contributions from different source regions in the four seasons are similar to those in Lu et al. (2002). We also found a good agreement (r=0.72, p<0.05) in the estimation of imported BC between this study and Lu et al. (2012) at several sites in the TP, although our estimates are higher than those from Lu et al. (2012). The back-trajectory approach modified in this study shows strong performance in identifying the BC source regions for the TP.

The simulated annual mean surface BC concentrations over the entire TP are shown in Figure 5a. The BC concentrations are high along the eastern and southern borders and low in the center of the TP. BC concentrations over the TP show strong spatial gradient, which is likely due to the blocking of BC transport by the mountains with high elevations (Cao et al., 2011; Zhao et al., 2017). Figures 5b-5f show the dominant source regions for the TP in the annual mean and by season. In the annual mean, 93% of surface BC in the TP comes from South Asia (47%) and East Asia (46%), which are two dominant source regions for the TP as BC aerosols from South Asia and East Asia can impact 64% and 34% areas of the TP, respectively (Figure 5b). Because of the leeward location of East Asia under prevailing westerlies, the influence of East Asia is constrained mainly in northern and eastern TP. In winter, South Asia is the dominant source region for 83% areas of the TP (Figure 5f), while in summer, East Asia is the dominant source region for 54% areas of the TP (Figure 5d).





We further divided the TP into five subregions, namely, eastern TP, southern TP, western TP, northern TP, and central TP (Figure 5a). The 20-year means in the subregions show different BC levels, seasonalities, and dominant BC source regions (Figures 5-7). Over the eastern TP, surface BC concentrations are the highest among the five subregions (Figure 6b). Over 75% of surface BC in the eastern TP is transported from East Asia (Figure 7b).

In the southern TP, surface BC concentrations are the 2nd highest among the five subregions, which are high in spring and low in the other seasons (Figure 6c). Such seasonality is likely resulted from the high fire emissions over South Asia in spring, the favorable atmospheric circulation for BC transport to the southern TP in spring, and the strong wet deposition of BC by the monsoon precipitation in summer (Chen et al., 2018). This seasonality is in consistency with observations in previous studies (Marinoni et al., 2010; Cong et al., 2015). South Asia is the dominant source region for surface BC in the southern TP year-round, with fractional contributions of over 85% (Figure 7c). The dominant contribution of South Asia to the southern TP was also suggested in previous studies (He et al., 2014a; Zhang et al., 2015; Yang et al., 2018). The second key source region for the TP is Southeast Asia.

Over the western TP, BC concentrations are the 3rd lowest among the five subregions, with a seasonality of high BC in winter and spring and low BC in summer and autumn (Figure 6d). The higher values in spring and winter agree with the BC measurements at sites in the western Himalayas (Nair et al., 2013). BC transport from South Asia contributes to 93% of surface BC in winter and 76% in summer (Figure 7d). Such seasonality with winter high and summer low in the fractional contribution of South Asia to surface BC over the western TP were also suggested by Zhang et al. (2015).

In the northern TP, BC concentrations are the 2nd lowest among the five subregions, which are at





maximum in winter and minimum in spring (Figure 6e). This seasonality is different from an

observational study, which reported that, over the Qilian Shan in the northern TP, surface BC

concentrations are highest in summer and lowest in autumn (Zhao et al., 2012). The dominant source

region for surface BC over the northern TP is South Asia in winter and East Asia in the other seasons

(Figure 7e). Influenced by the prevailing westerlies, source regions west of the TP (central Asia, the

region of other Asia, Europe, and Africa in Figure 1b) contribute to surface BC in the western and

northern TP more than to other TP subregions (Figure 7e). Source regions west of the TP can contribute

to 5-11% and 9-17% surface BC in the western and northern TP, respectively (Figures 7d and 7e).

Among the five subregions, the central TP is with the lowest BC concentrations (Figure 6f). Source

regions other than South Asia and East Asia contribute only 5% or less to surface BC in the central TP

(Figure 7f). Seasonally, BC concentrations over the central TP is higher in spring and winter than in

summer and autumn. This simulated seasonality is different from the observed one at a site in the

central TP reported in Wang et al. (2016), who showed that BC concentrations are higher in spring and

lower in winter from November 2012 to June 2013. In spring and winter, South Asia respectively

contributes to 72% and 91% surface BC in this subregion. In contract, East Asia contributes to 58%

surface BC there in summer.

**4 Interannual variation in surface black carbon over the Tibetan Plateau**

**4.1 Influences of biomass burning on surface black carbon over the Tibetan Plateau**

Figure 8 shows the anomalies of surface BC concentrations averaged over the TP from the three GEOS-

Chem simulations. The simulations from CTRL and FixBB are significantly correlated with each other

in all the seasons (r=0.45, p<0.05 in spring, r>0.8, p<0.05 in the other three seasons), indicating the

important role of meteorology in the interannual variation in surface BC concentrations in the TP.





Remarkably, in spring (Figure 8a), the correlation coefficient of BC anomalies between CTRL and

FixMet simulations reaches 0.87 (p<0.05), indicating the importance of biomass burning to the

interannual variation in BC in spring. The largest anomaly of BC concentrations from CTRL simulation

is in 1999. The comparison between CTRL and FixMet simulations suggests that this strong anomaly is

largely explained by biomass burning. Even if we exclude the extreme year 1999, the correlation

(r=0.77, p<0.05) between CTRL and FixMet simulations remains significant, indicating the strong

influence of biomass burning on the variation in surface BC from year to year.

To further examine influence of biomass burning in spring, we integrally analyzed data from ATSR

satellite fire counts, GFED3 fire emissions, and the GEOS-Chem simulations. Both ATSR and GFED3

data show that fires occur frequently over the Indo-Gangetic Plain, central India, and Southeast Asia

(Figures 1 and 9). Fire activities in Asia are well described in the GFED3 inventory that is used in the

GEOS-Chem simulations (Figures 9b and 9c). We found the interannual variation in BC anomalies in

the TP from CTRL simulation is significantly correlated to the fire counts in the Indo-Gangetic Plain

(r=0.76, p<0.05), and central India (r=0.67, p<0.05). The correlation was insignificant for Southeast

Asia (r=0.19, p>0.05). In spring of 1999, extreme fire activities occurred in the Indo-Gangetic Plain and

central India (Figure 9b). Driven by the favourable atmospheric circulation, the strong BC emissions

from the extremely active fires greatly enhanced surface BC concentrations in the TP (Figure 9d). In the

CTRL simulation, positive BC anomalies appear over the entire TP, with a regional mean of 0.15 $\mu$g m$^{-3}$

or 31% relative to the 1995-2014 climatology (Figure 8a). Additionally, in winter, biomass burning was

extremely strong in 1998 (Figure 8d). The extremely active fires enhanced the regional mean surface

BC concentrations in the TP by 0.02 $\mu$g m$^{-3}$ or 5% relative to the climatology.

According to Figure 8, the simulations between CTRL and FixMet are not significantly correlated





in summer, autumn, and winter, suggesting that meteorology plays an important role in modulating the interannual variation in surface BC in the TP. Such a role will be explored in sections 4.2 and 4.3 from

360 the influences of the Asian monsoon on BC transport to the TP in summer and winter, respectively.

## 4.2 Influences of the Asian summer monsoon on the transport of black carbon to the Tibetan Plateau

The TP can largely impact the Asian monsoon system through thermal and dynamic processes (Wu et al.,

365 2015). In the meantime, the Asian monsoon can significantly influence the transport of atmospheric species to the TP (Xu et al., 2014). In this section, we show the influences of two Asian monsoon subsystems on the interannual variation in BC transport to the TP in summer. We employed a unified dynamical monsoon index to represent the strength of East Asian summer monsoon (EASM) and South Asian summer monsoon (SASM). The index was proposed by Li and Zeng (2002) and it has been

370 widely applied to quantify the impact of the Asian monsoon on air pollutants in Asia (Mao et al., 2017; Lu et al., 2018). Using this index, Han et al. (2019) found a close correlation between the EASM and ozone transport from foreign regions to East Asia. The calculation of the index was introduced in Li and Zeng (2002) and Han et al. (2019). The index is respectively termed as EASM index (EASMI) and SASM index (SASMI), when it is applied to represent the strength of EASM and SASM. A higher

375 EASMI indicates a stronger EASM and a higher SASMI indicates a stronger SASM.

  Figure 10a shows the spatial distribution of the correlation between BC transport to the TP and wind at 850 hPa at each of the grids in summer. As known from Figure 4b, central China is a dominant source region for the TP in summer, accounting for 63% of the imported BC from East Asia to the TP

380 surface. In Figure 10a, BC from central China correlates significantly with the zonal wind at 850 hPa in central China (Figure 10a), with a regional mean correlation coefficient of -0.55 (p<0.05). Westward



winds (negative in the zonal component of wind vector) over central China favor BC transport from

East Asia to the TP. Furthermore, the EASMI also correlates negatively with the zonal wind at 850 hPa

over central China (Figure 10b). When the EASM is stronger, the zonal wind in the monsoon circulation

weakens over this region (Yang et al., 2014; Han et al., 2019), suggesting that westward winds may

occur more or with higher speed. Therefore, BC transport to the TP from central China is enhanced

(Figure 10c), as a significantly positive correlation is found between the strength of the EASM and BC

transport from central China to the TP surface (r=0.49, p<0.05) and between the strength of the EASM

and BC transport from central China to the eastern TP surface (r=0.48, p<0.05). This is further

confirmed by the differences in BC transport to the TP surface between summers with strong and weak

EASM (Figure 10d).

   How the SASM impacts the BC transport from South Asia over the TP surface in summer is also

examined (Figure 11). Serving as a heat source in the Asian summer monsoon system, the TP promotes

strong convection and modulates the meridional circulation (Xu et al., 2014). Driven by the meridional

circulation, BC in South Asia can be transported northward and upward to the TP. BC transport from

northeastern South Asia to the TP accounts for 30% of the total BC transport from South Asia (Figure

4b). Interannually, BC transport from northeastern South Asia is significantly correlated with the

meridional wind at 500 hPa (r=0.65, p<0.05, Figure 11a), which is also closely correlated to the strength

of the SASM (Figure 11b). In strong SASM years, an anomalous cyclone locates over the northern

South Asia at 500 hPa and correspondingly the meridional wind over the northeastern South Asia is

increased (Figure 11d). This well explains why the interannual variation in BC transport from

northeastern South Asia correlates positively with the strength of the SASM (r=0.55, p<0.05 for the TP,

r=0.56, p<0.05 for the STP, Figure 11c). Among all source regions, the differences in BC transport from

northeastern South Asia to the TP is largest between summers with strong and weak EASM (Figure 11d).





## 4.3 Influences of the Asian winter monsoon on the transport of black carbon to the Tibetan Plateau

The Asian winter monsoon is a predominant climate feature in Asia and an important modulator of the

distribution and transport of air pollutants (Mao et al., 2017; Zhu et al., 2017). However, the impact of

the Asian winter monsoon on the interannual variation in BC transport to the TP scantly studied. In this

section, we assess such impact with two climate indices. We measure the intensity of East Asian winter

monsoon (EAWM) by an index defined by Jhun and Lee (2004). The EAWM index (EAWMI)

represents the EAWM intensity by the meridional wind shear associated with the jet stream in the upper

troposphere. It can be calculated by the difference in the regional averaged zonal wind speed at 300 hPa

between the areas 27.5-37.5°N, 110-170°E and 50-60°N, 80-140°E. Using the EAWMI, it is found that

the EAWM is closely correlated with the interannual variation in pollution transport over East Asia (Li

et al., 2016; Han et al., 2019). Furthermore, the Siberian High is a key component of the EAWM system

(Wu and Wang, 2002) and its strength can be described using an index defined by Wu and Wang (2002).

This Siberian High index (SHI) can be calculated from the regional mean sea level pressure over the

area of the Siberian High (40-60°N, 80-120°E). The EAWMI and SHI are highly correlated (r=0.72,

p<0.05).

Figure 12 illustrates a connection between the EAWM and BC transport from East Asia to the TP

surface. We mainly focused on BC transport from central China, as this area contributes to 54% of the

total BC transport from East Asia to the TP (Figure 4d). BC transport from central China to the TP

surface layer correlates significantly with the zonal wind at 850 hPa over central China (r=-0.73, p<0.05,

Figure 12a). The zonal wind over this region is also correlated with the strength of the EAWM (r=-0.5,

p<0.05) and the strength of the Siberian High (r=-0.65, p<0.05) (Figure 12b).  A significant correlation



(r=0.59, p<0.05) is found between the strength of the EAWM and BC transport from central China to

the TP surface (Figure 12c). When the EAWM is stronger, the more frequent or stronger westward

winds can enhance BC transport from central China to the TP (Figure 12d).

A connection between BC transport from South Asia to the TP surface and the Siberian High is

shown in Figure 13 for winter. BC over the northern South Asia can be transported efficiently to the TP

by the prevailing subtropical westerlies. Northern South Asia contributes to 70% of BC transported

from South Asia to the TP surface (Figure 4d). The contribution of northern South Asia to surface BC in

the TP is significantly related (r=0.72, p<0.05) to the zonal wind at 500 hPa over the TP (Figure 13a).

The westerlies over the TP are also correlated with the strength of Siberian High (Figure 13b). The

elevated zonal wind in the middle troposphere over the TP in winters with strong Siberian High can

enhance the BC transport from northern South Asia to the TP (Figure 13d). Significant correlations are

found between the strength of the Siberian High and BC transport from northern South Asia to the TP

surface (r=0.66, p<0.05) and between the strength of the Siberian High and BC transport from northern

South Asia to the southern TP surface (r=0.65, p<0.05) (Figure 13c). In addition, the contribution of

northern South Asia to surface BC in the western TP increases significantly with the meridional wind at

500 hPa over the western TP (r=0.64, p<0.05). The differences in BC transport from northern South

Asia to the TP is largest between winters with strong and weak SHI (Figure 13d).

## 5 Discussion and conclusions

Using a global chemical transport model, GEOS-Chem, we characterized the variation in surface BC

over the TP in 20 years from 1995 to 2014. By comparing with observations available in the literature,

GEOS-Chem simulations show good performance in reproducing the spatial distribution, magnitude,

and seasonal variation in surface BC over the TP. Applying an approach that combines the BC



simulations from GEOS-Chem and backward trajectories from HYSPLIT, we identified the source

regions for surface BC in the TP and demonstrated the influences of atmospheric transport and biomass

burning on the interannual variation in surface BC over the TP. The major conclusions, along with

discussion, are drawn as follows.

Based on the 20-year mean, surface BC in the TP is mainly influenced by two source regions: East

Asia and South Asia. The influence of East Asia is dominant in summer while the influence of South

Asia is dominant in winter. We divided the TP into five subregions: eastern, southern, western, northern,

and central TP. Surface BC concentrations are higher in the southern and eastern TP than in the other

subregions. Surface BC in the southern and eastern TP comes mainly from South Asia and East Asia,

respectively. Over the western TP, surface BC comes mainly from South Asia. Over the northern TP, the

dominant BC source region is South Asia in winter and East Asia in the other seasons. Over the central

TP, the dominant BC source region is East Asia in summer and South Asia in the other seasons.

Interannually, from 1995 to 2014, biomass burning can explain over 75% of the variation in

springtime surface BC concentrations over the TP if biomass burning and meteorology are both

considered in GEOS-Chem simulations. Indeed, springtime surface BC in the TP is significantly

correlated to the total number of fire counts over the Indo-Gangetic Plain in South Asia (r=0.76, p<0.05),

according to ATSR satellite data. In the spring of 1999, the extremely strong biomass burning in South

Asia largely elevated surface BC concentrations (0.15 $\mu$g m$^{-3}$ or 31% relative to the climatology) over

the TP. We noticed that the strong biomass burning in South Asia in the winter of 1998 also enhanced

BC concentrations over the TP.

The interannual variation in surface BC over the TP are greatly influenced by meteorology.



Specifically, the Asian monsoon system alters the long-range transport of BC to the TP by modulating the atmospheric circulation. In summer, when the EASM is stronger, the more frequent or stronger

westward wind in the lower troposphere can enhance BC transport from central China to the TP. When the SASM is stronger, the increased meridional wind over the northeastern South Asia in the middle troposphere can enhance BC transport from northeastern South Asia to the TP. In winter, when the EAWM is stronger, the reduced zonal wind in the lower troposphere tends to increase BC transport from central China to the TP. A stronger Siberian High can enhance the zonal wind in the middle troposphere

over the TP and consequently increases BC transport from northern South Asia to the TP.

The findings in this study provide an enhanced understanding of the long-range transport of BC to the TP. We comprehensively assessed the BC transport from worldwide source regions to the TP. Our results reveal the source regions of surface BC over the entire TP in the four seasons, which was

investigated by limited studies (Zhang et al., 2015). The influences of South Asia and East Asia on the TP were noticed by previous studies. Most of them were focused on limited locations (Cao et al., 2011; Engling et al., 2011; Chen et al., 2018) or in one or few seasons (Zhao et al., 2017; Wang et al., 2018). Here, we further quantified the influence of South Asia and East Asia over the entire TP in the four seasons, in terms of both fractional contribution and affected areas in the TP. Moreover, we identified

three key areas within South Asia and East Asia and found that the contribution of BC from there to surface BC in the TP is highest among South Asia and East Asia.

Biomass burning is an important source of atmospheric BC in the TP (Zhang et al., 2015). It was observed that BC emissions from biomass burning in South Asia could be transported to the TP by the

atmospheric circulation (Cong et al., 2015), and resulted in high BC episodes in the southern TP (Engling et al., 2011; Putero et al., 2014). Only limited numerical studies explored connections between





biomass burning and surface BC in the TP over a long-term period (Mao and Liao, 2016). Here, we demonstrated that biomass burning is an important driver of the interannual variation in surface BC over the TP in spring. In particular, we found that there were extremely strong fire activities over Indo-

Gangetic Plain, central India, and Southeast Asia from 1998 winter to 1999 spring that largely enhanced surface BC concentrations over the entire TP. This extreme anomaly in fire activities and associated influence on BC over the TP may have not been fully documented.

We found that the Asian monsoon system can significantly modulate the interannual variation in

BC transport from South Asia and East Asia to the TP. Asian monsoon can influence the atmospheric circulation over the TP and its surroundings (Xu et al., 2014; Han et al., 2019). For summer, previous studies mainly focused on the transport pathway build by the SASM, which can meridionally transport BC from South Asia to the TP (Zhao et al., 2017; Kang et al., 2019). In this study, we further revealed that the EASM can modulate the westward transport of BC from central China to the TP. In winter, the

Asian monsoon system also significantly influences the BC transport from northern South Asia and central China to the TP. These results can shed some light on the transport mechanisms of other atmospheric species to the TP, such as water vapour.

Numerical simulations, including backward-trajectory (Ming et al., 2009, 2010; Lu et al., 2012),

adjoint (Kopacz et al., 2011), and tagging tracer (Zhang et al., 2015) simulations, have been used to identify BC transport from sources to the TP. Modifying an approach proposed by Lu et al. (2012), we developed an efficient and stable method which shows strong performance in revealing the factional contribution of BC transport from different source regions to the TP by season and year. This method can explicitly show the spatial distribution of the contribution of different source regions to surface BC

over the TP. It is feasible for users of a chemical transport model to estimate BC transport from different





source regions to a receptor region if adjoint and tagged modes of the model are unavailable to them.

This study is subject to some limitations. Numerical simulations have advantages of covering large

areas over the entire TP and long periods, such as 20 years in this study. GEOS-Chem has been used in

multiple studies on BC aerosols over the TP (Kopacz et al., 2011; Lu et al., 2012; He et al., 2014b; Mao

and Liao. 2016). However, according to this and earlier studies, there are discrepancies between

observations and simulations from GEOS-Chem. He et al. (2014b) suggested that BC aerosols over the

TP may be underestimated by GEOS-Chem. A smaller seasonal variation in BC concentrations from

simulations than from observations is also revealed in this study. All of these imply uncertainty in

simulating the absolute BC concentrations over the TP by GEOS-Chem. Additionally, the simulations

are with a resolution of 2° latitude by 2° longitude. Such a resolution may not fully capture processes in

the sub-grid scale, such as the mountain-valley wind (Cong et al., 2015). Using regional models at

higher resolutions in the future can better describe the terrain effect in the TP. This study is focused on

natural drivers (biomass burning and meteorology) that are connective to the interannual variation in

BC over the TP. The interannual anthropogenic emissions warrant further studies.

**Data availability**

The GEOE-Chem model is publicly available at http://geos-chem.org. The HYSPLIT model can be

acquired from http://www.arl.noaa.gov/HYSPLIT_info.php. The meteorological and fire data were

download from https://www.esrl.noaa.gov/psd/data/gridded/data.ncep.reanalysis.surface.html and

https://earth.esa.int/web/guest/home, respectively.

**Author contributions**

JL, YW, and HH designed the research. HH and YW performed the study. HH, YW, YL, HC, and YZ



analyzed the data. HH, YW, and JL wrote the paper. TZ, BZ, HL, QW, SL, TW, MX, and ML

contributed insight and comments.

## Competing interests

The authors declare no conflict of interest.


## Acknowledgments

We are grateful to the following model and data providers. The GEOE-Chem model is developed and

managed by the Atmospheric Chemistry Modeling Group at Harvard University. The HYSPLIT model

is developed by NOAA Air Resources Laboratory. The meteorological and fire data were respectively

acquired from NOAA Earth System Research Laboratory and European Space Agency.

## Financial support

This research is supported by the Chinese Ministry of Science and Technology National Key R&D

Program of China (grant nos. 2019YFA0606803 and 2016YFA0600204) and the Natural Science

Foundation of China (grant nos. 91744209, 91544230, and 41375140).

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

        Sources and physicochemical characteristics of black carbon aerosol from the southeastern Tibetan

©c Author(s) 2020. CC BY 4.0 License.





Plateau: internal mixing enhances light absorption, Atmos. Chem. Phys., 18, 4639-4656,
https://doi.org/10.5194/acp-18-4639-2018, 2018.

Wu, B., and Wang, J.: Winter Arctic Oscillation, Siberian High and East Asian winter monsoon,
Geophys. Res. Lett., 29, 3-1-3-4, https://doi.org/10.1029/2002GL015373, 2002.

Wu, G., Duan, A., Liu, Y., Mao, J., Ren, R., Bao, Q., He, B., Liu, B., and Hu, W.: Tibetan Plateau
climate dynamics: recent research progress and outlook, Natl. Sci. Rev., 2, 100-116,
https://doi.org/10.1093/nsr/nwu045, 2015.

Xu, B., Cao, J., Hansen, J., Yao, T., Joswia, D. R., Wang, N., Wu, G., Wang, M., Zhao, H., Yang, W., Liu,
X., and He, J.: Black soot and the survival of Tibetan glaciers, P. Natl. Acad. Sci. USA, 106,
22114-22118, https://doi.org/10.1073/pnas.0910444106, 2009.

Xu, X., Zhao, T., Lu, C., Guo, Y., Chen, B., Liu, R., Li, Y., and Shi, X.: An important mechanism
sustaining the atmospheric "water tower" over the Tibetan Plateau, Atmos. Chem. Phys., 14,
11287-11295, https://doi.org/10.5194/acp-14-11287-2014, 2014.

Xu, Y., Ramanathan, V., and Washington, W. M.: Observed high-altitude warming and snow cover
retreat over Tibet and the Himalayas enhanced by black carbon aerosols, Atmos. Chem. Phys., 16,
1303-1315, https://doi.org/10.5194/acp-16-1303-2016, 2016.

Yang, J., Kang, S., Ji, Z., and Chen, D.: Modeling the origin of anthropogenic black carbon and its
climatic effect over the Tibetan Plateau and surrounding regions, J. Geophys. Res.-Atmos., 123,
671-692, https://doi.org/10.1002/2017JD027282, 2018.

Yang, Y., Liao, H., and Li, J.: Impacts of the East Asian summer monsoon on interannual variations of
summertime surface-layer ozone concentrations over China, Atmos. Chem. Phys., 14, 6867-6879,
https://doi.org/10.5194/acp-14-6867-2014, 2014.

Zhang, R., Wang, H., Qian, Y., Rasch, P. J., Easter, R. C., Ma, P.-L., Singh, B., Huang, J., and Fu, Q.:
Quantifying sources, transport, deposition, and radiative forcing of black carbon over the





Himalayas and Tibetan Plateau, Atmos. Chem. Phys., 15, 6205-6223, https://doi.org/10.5194/acp-15-6205-2015, 2015.

Zhang, X. Y., Wang, Y. Q., Zhang, X. C., Guo, W., and Gong, S. L.: Carbonaceous aerosol composition over various regions of China during 2006, J. Geophys. Res.-Atmos., 113, D14111, https://doi.org/10.1029/2007JD009525, 2008.

Zhang, Y., Kang, S., Sprenger, M., Cong, Z., Gao, T., Li, C., Tao, S., Li, X., Zhong, X., Xu, M., Meng, W., Neupane, B., Qin, X., and Sillanpää, M.: Black carbon and mineral dust in snow cover on the
Tibetan Plateau, The Cryosphere, 12, 413-431, https://doi.org/10.5194/tc-12-413-2018, 2018.

Zhao, S., Ming, J., Xiao, C., Sun, W., and Qin, X.: A preliminary study on measurements of black carbon in the atmosphere of northwest Qilian Shan, J. Environ. Sci., 24, 152-159, https://doi.org/10.1016/S1001-0742(11)60739-0, 2012.

Zhao, S., Tie, X., Long, X., and Cao, J.: Impacts of Himalayas on black carbon over the Tibetan Plateau
during summer monsoon, Sci. Total Environ., 598, 307-318, https://doi.org/10.1016/j.scitotenv.2017.04.101, 2017.

Zhu, Y., Liu, J., Wang, T., Zhuang, B., Han, H., Wang, H., Chang, Y., and Ding, K.: The impacts of meteorology on the seasonal and interannual variabilities of ozone transport from North America to East Asia, J. Geophys. Res., 122, 10612–10636, https://doi.org/10.1002/2017JD026761, 2017.

Zhuang, B. L., Li, S., Wang, T. J., Liu, J., Chen, H. M., Chen, P. L., Li, M. M., and Xie, M.: Interaction between the Black Carbon Aerosol Warming Effect and East Asian Monsoon Using RegCM4, J. Climate, 31, 9367-9388, https://doi.org/10.1175/JCLI-D-17-0767.1, 2018.



Table 1. Comparison between observed and simulated annual mean surface BC concentrations (in µg m⁻³) during the corresponding periods at various urban, rural, and remote sites (see Figure 1a) over the TP and its surrounding regions.

| Site category | Site name | Latitude (ºN) | Longitude (ºE) | Altitude (m) | Time period | Observed BC (µg m⁻³) | Simulated BC (µg m⁻³) | References |
|---|---|---|---|---|---|---|---|---|
| Remote | Nagarkot | 27.7 | 85.5 | 2150 | 1999-2000 | 1.0 | 1.26 | Carrico et al. (2003) |
| | NCOP | 28.0 | 86.8 | 5079 | 2006 | 0.2 | 0.52 | Bonasoni et al. (2010) |
| | Manora Peak | 29.4 | 79.5 | 1950 | 2006 | 1.13 | 0.86 | Ram et al. (2010a) |
| | NCOS | 30.8 | 91.0 | 4730 | 2006 | 0.14 | 0.19 | Ming et al. (2010) |
| | Langtang | 28.1 | 85.6 | 3920 | 1999-2000 | 0.41 | 0.6 | Carrico et al. (2003) |
| | Zhuzhang | 28.0 | 99.7 | 3583 | 2004-2005 | 0.35 | 0.26 | Qu et al. (2008) |
| Rural | Kharagpur | 22.5 | 87.5 | 28 | 2006 | 5.56 | 4.45 | Nair et al. (2012) |
| | Kanpur | 26.4 | 80.3 | 142 | 2006 | 3.77 | 2.48 | Ram et al. (2010b) |
| | Gandhi College | 25.9 | 84.1 | 158 | 2006 | 4.88 | 4.05 | Ganguly et al. (2009) |
| Urban | Delhi | 28.6 | 77.2 | 260 | 2006 | 13.59 | 2.92 | Beegum et al. (2009) |
| | Dibrugarh | 27.3 | 94.6 | 111 | 2008-2009 | 8.91 | 1.13 | Pathak et al. (2010) |
| | Lhasa | 29.7 | 91.1 | 3663 | 2006 | 3.711 | 0.19 | Zhang et al. (2008) |
| | Dunhuang | 40.2 | 94.7 | 1139 | 2006 | 4.111 | 0.18 | Zhang et al. (2008) |




Figure 1. (a) Annual BC emissions from biomass burning averaged over 1997-2011 from GFED3. Red

triangles (for Table 1) and blue dots (for Table 1 and Figure 2) indicate the locations of the observation

sites used for model evaluation. (b) Source regions defined for the estimation of BC transport to the TP.

The abbreviations are for central Asia (CAS), East Asia (EAS), South Asia (SAS), Southeast Asia

(SEAS), and the region of other Asia, Europe, and Africa (OAEA). The dark black line in (a) encloses

the domain of the TP, corresponding the white areas in (b).






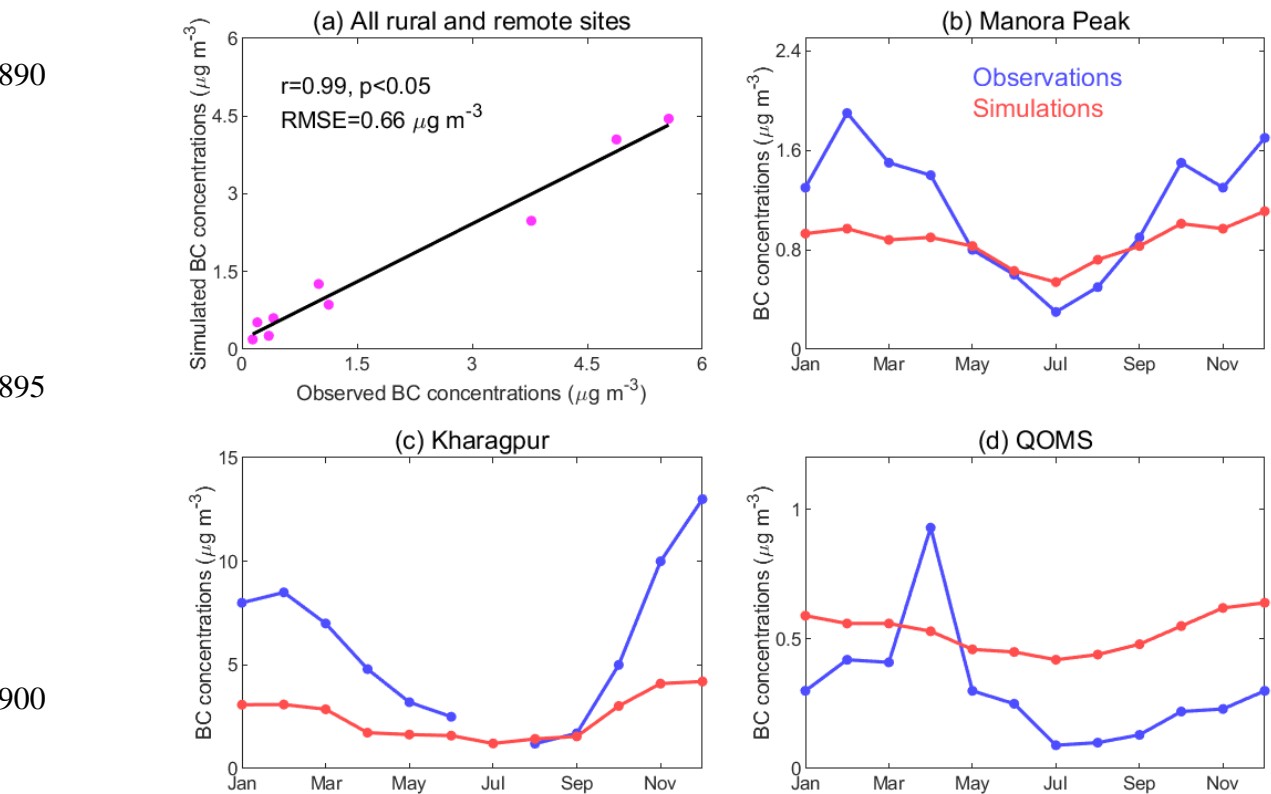

Figure 2. (a) Annual means of the observed and simulated surface BC concentrations at the rural and remote sites (see Table 1) in the TP and its surrounding regions in 2006. Comparisons between the monthly observations and simulations at (b) Manora Peak, (c) Kharagpur, and (d) QOMS (blue dots in Figure 1a) in 2006. Correlation coefficient ($r$) and root mean square error (RMSE) between the observed and simulated BC at all the rural and remote sites are also shown in (a). Observations at QOMS (28.36°N, 86.95°E, 4276 m) are from Chen et al. (2018).

Figure 3. An example of the estimation of BC transport to the TP surface in April 2005. (a) Surface BC concentrations (in µg m$^{-3}$). (b) 7-day backward trajectories (in meters above the ground) arriving at the TP surface. (c) The amount of BC transported to the TP surface from source regions (in µg m$^{-2}$ d$^{-1}$). The dark black line encloses the domain of the TP.







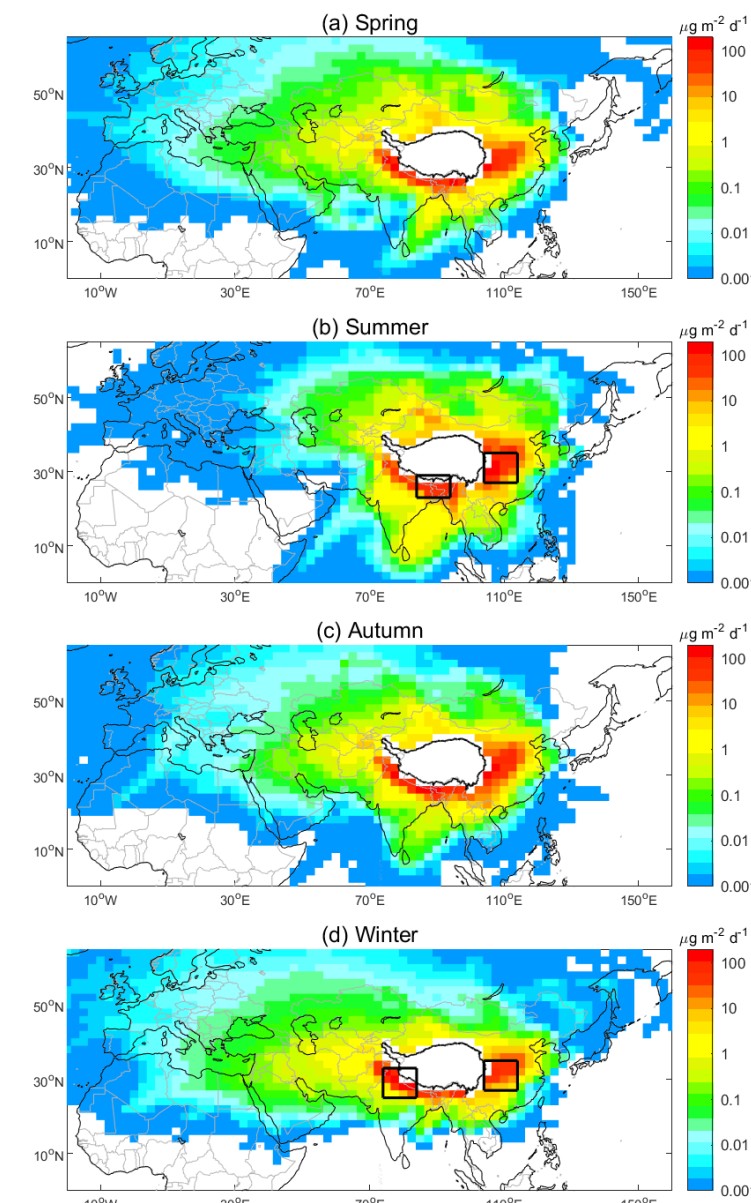

Figure 4. The amount of BC transported to the TP surface (in µg m⁻² d⁻¹) from source regions in (a) spring, (b) summer, (c) autumn, and (d) winter. The values are based on 20-year (1995-2014) means from CTRL simulation. The boxed areas in (b) indicate central China (CCH; 27-35ºN, 104-114ºE) and northeastern South Asia (NESAS; 23-29ºN, 84-94ºE). The boxed areas in (d) indicate CCH and northern South Asia (NSAS; 25-33ºN, 74-84ºE). The dark black line encloses the domain of the TP.

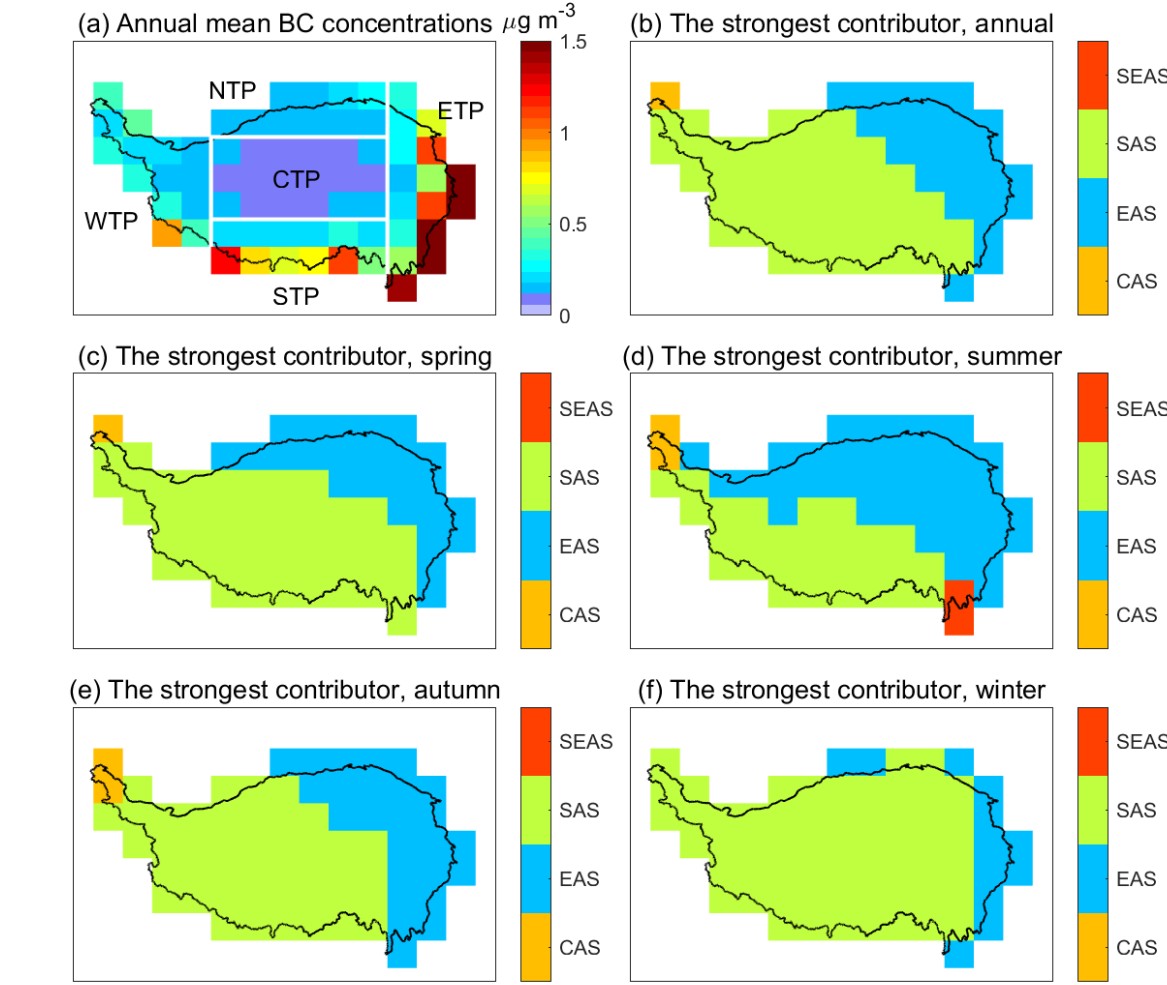

Figure 5. Surface BC over the TP and the dominant BC source regions for the TP. (a) Annual mean BC concentrations over the TP. (b-f) The dominant BC source regions (Figure 1b) for each model grid in the TP for (b) the annual mean, (c) spring, (d) summer, (e) autumn, and (f) winter. The values are 20-year mean (1995-2014) from CTRL simulation. Source regions in (b)-(f) include central Asia (CAS), East Asia (EAS), South Asia (SAS), and Southeast Asia (SEAS). The white solid lines in (a) separate the subregions of the TP defined in this study, i.e., western TP (WTP), northern TP (NTP), central TP (CTP), southern TP (STP), and eastern TP (ETP). The dark black line encloses the domain of the TP.





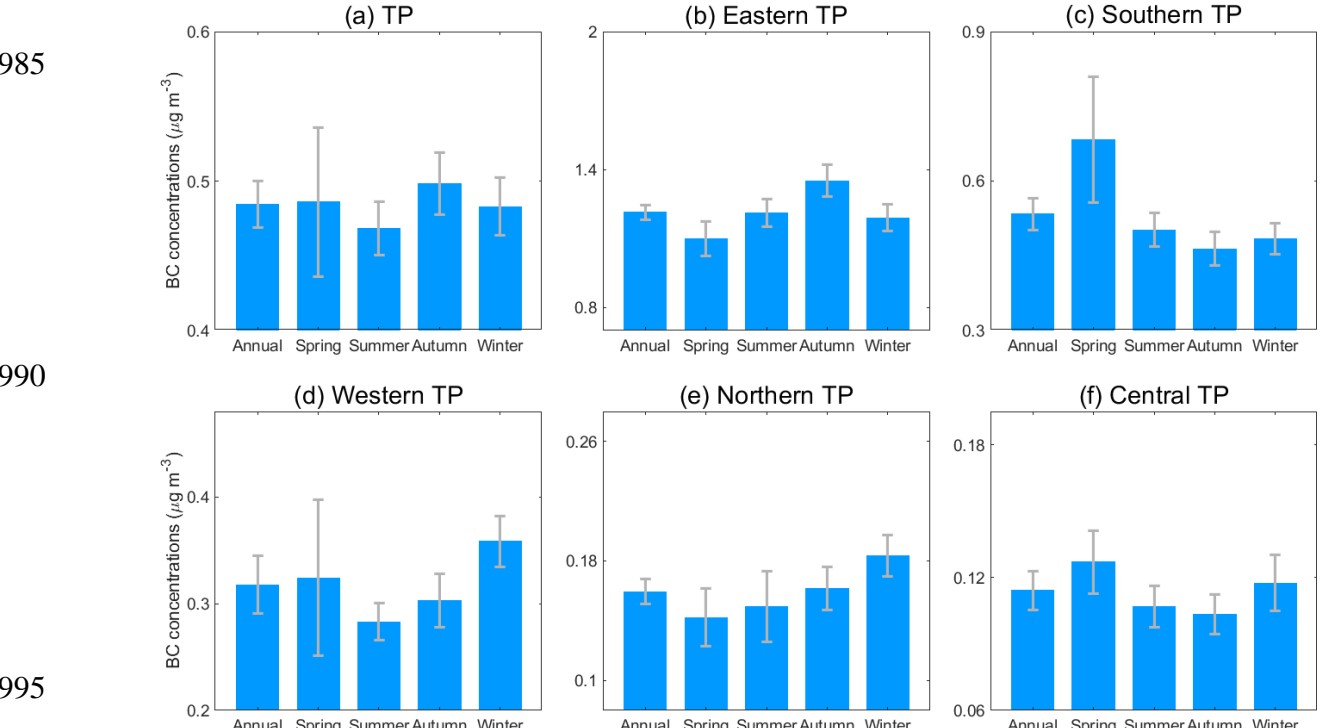

Figure 6. Seasonal variations of surface BC concentrations averaged over the TP and its subregions. The values are 20-year (1995 -2014) means from CTRL simulation. The error bars indicate the standard deviation. Note the y-scales are different for different subregions, ranging from the largest in eastern TP to the smallest in central TP.



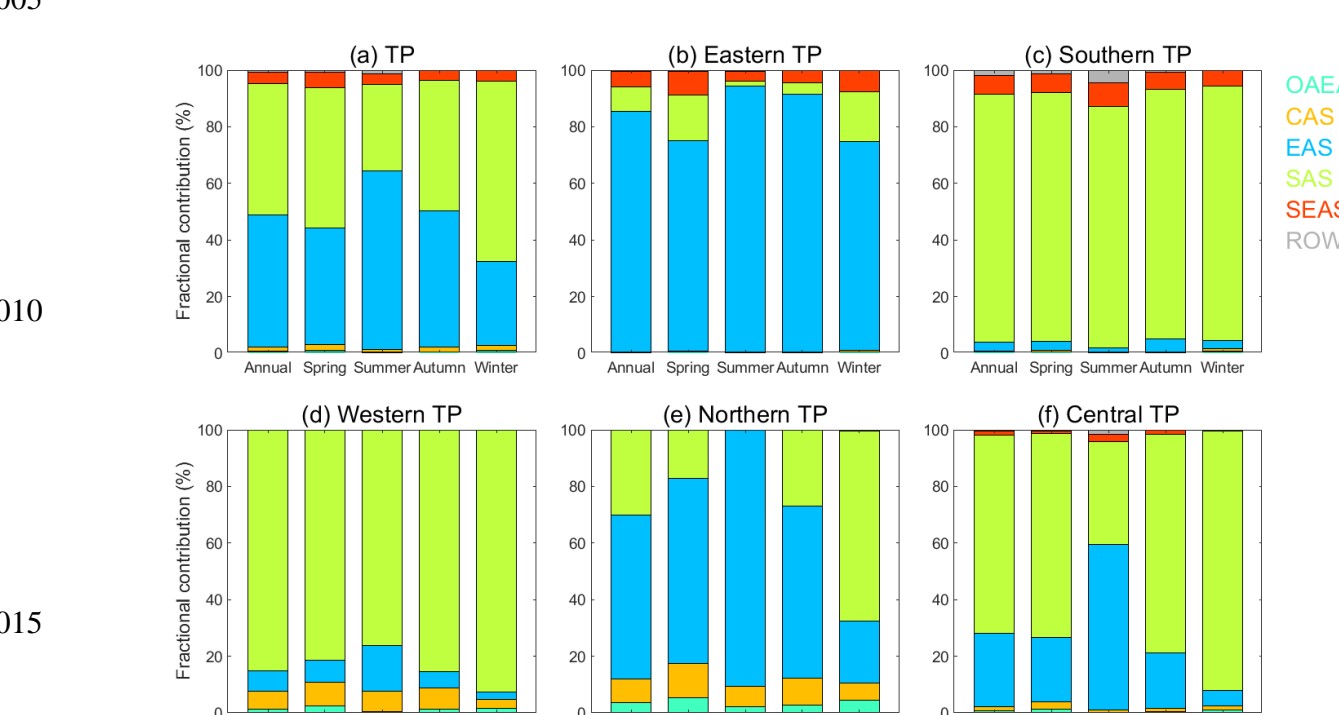

Figure 7. Fractional contributions of different source regions to surface BC over the TP and its

subregions. The abbreviations are for central Asia (CAS), East Asia (EAS), South Asia (SAS), Southeast

Asia (SEAS), the region of other Asia, Europe, and Africa (OAEA), and rest of the world (ROW).





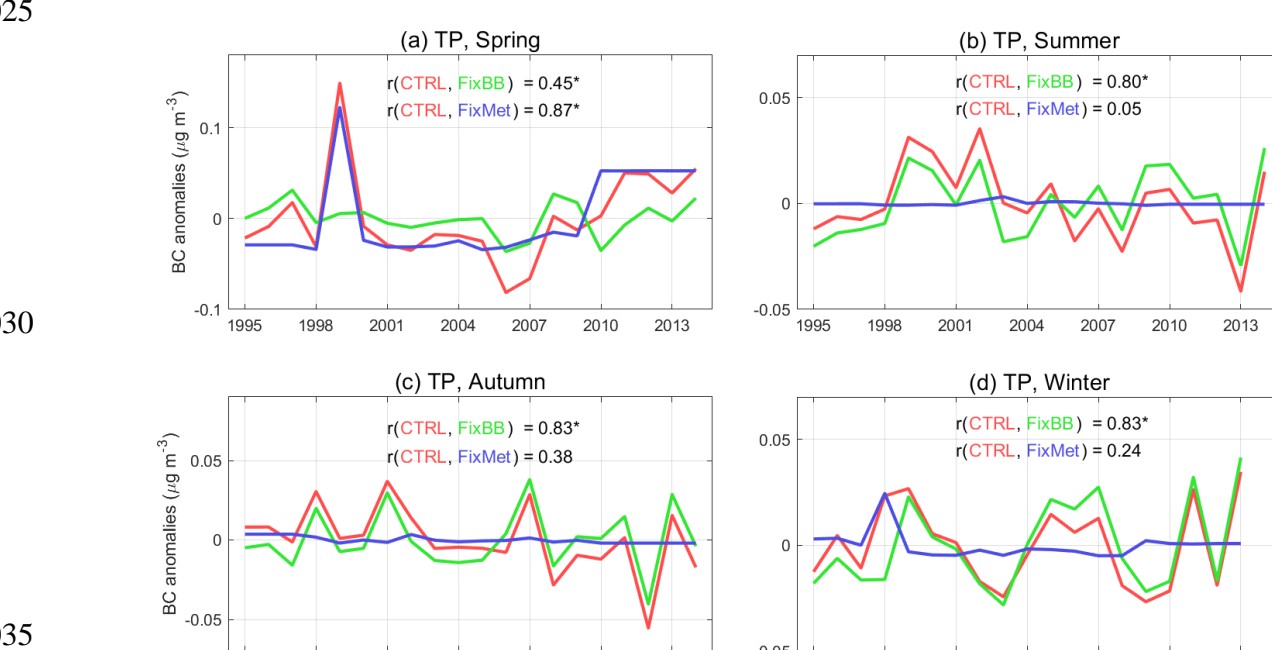

Figure 8. Interannual variations of the anomalies of surface BC concentrations averaged over the TP from different simulations during 1995-2014 in the four seasons. Red, green, and blue lines indicate CTRL, FixBB, and FixMet simulations, respectively. The anomalies are the BC concentrations in a given year minus those in the 20-year mean. A correlation coefficient (*r*) with '*' indicates that the *r* is statistically significant ($p<0.05$).



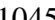

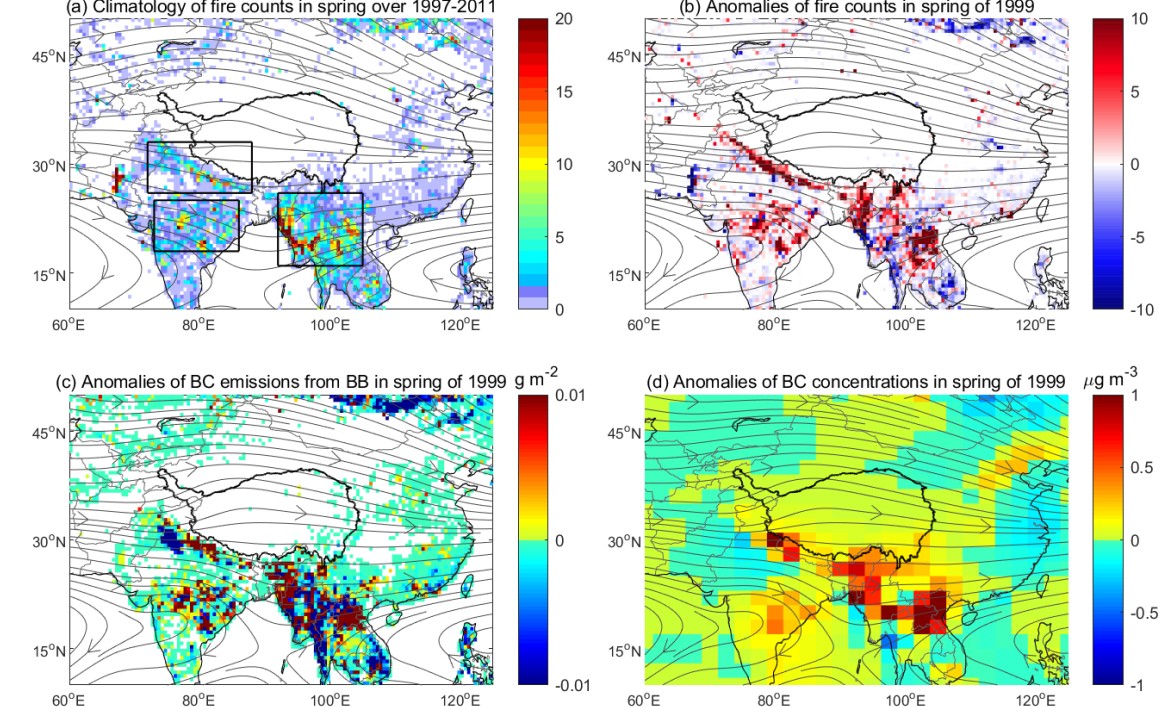

Figure 9. Extremely strong biomass burning in spring of 1999 and the corresponding surface BC
anomalies and wind field over the TP and surrounding regions. (a) Climatology of fire counts from
ATSR in spring overlaid with streamlines at 500 hPa over 1997-2011. (b) Anomalies of fire counts from
ATSR overlaid with anomalous streamlines at 500 hPa in spring of 1999. (c) The same as (b), but for
the anomalies of BC emissions from biomass burning (BB) from GFED3. (d) The same as (b), but for
the anomalies of BC concentrations from CTRL simulation. The anomaly in (b)-(d) is the corresponding
value in 1999 minus that over 1997-2011. The boxed areas in (a) indicate the fire regions in the Indo-
Gangetic Plain, central India, and Southeast Asia. The dark black line encloses the domain of the TP.

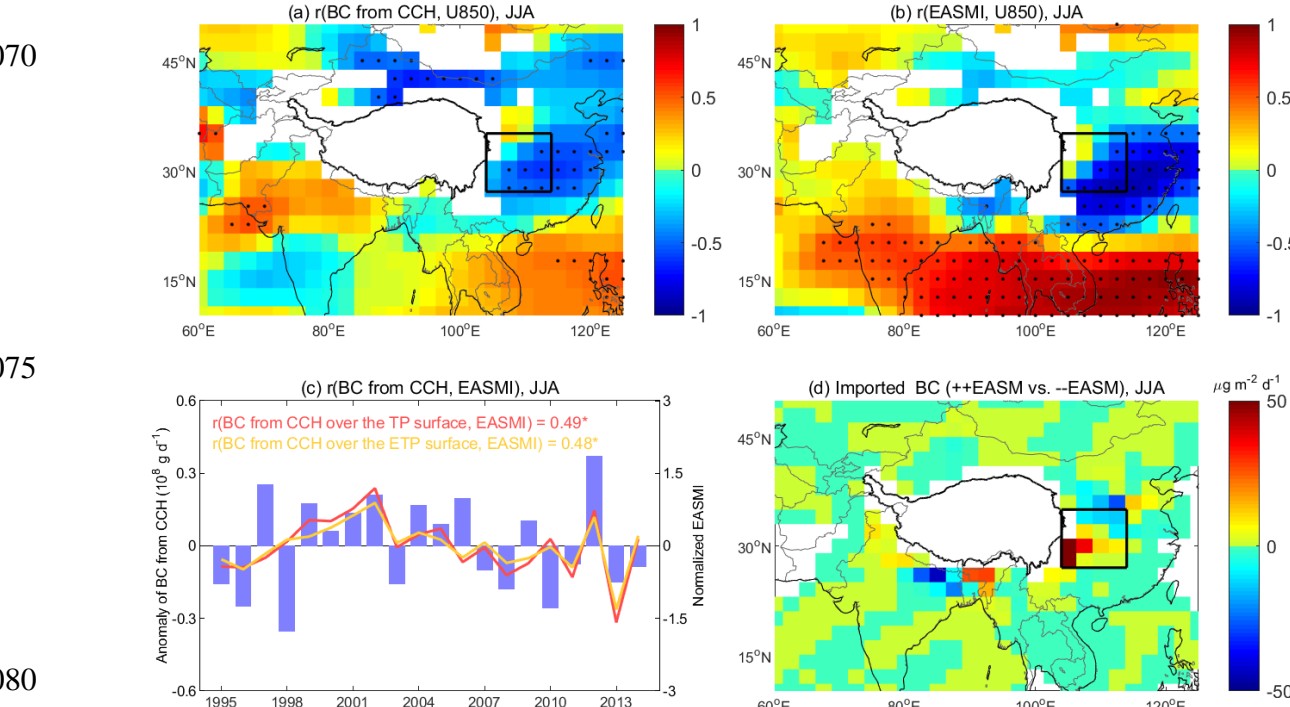

Figure 10. Connection between BC transport from East Asia to the TP surface and the EASM in summer. (a) Correlation coefficients (*r*) between the zonal wind at 850 hPa (U850) and imported BC from central China (CCH) over the TP surface. (b) *r* between the EASMI and U850. (c) Interannual variations in the intensity of the EASM, and imported BC from CCH to the surface of the TP and eastern TP (ETP). (d) Differences in BC transport to the TP surface between the years with strong and weak EASM. Dots in (a) and (b) indicate the *r* in the corresponding grid is statistically significant ($p<0.05$). The unfilled grids in (a) and (b) are due to the topography. A *r* with '*' in (c) indicates that the *r* is statistically significant ($p<0.05$). Boxed areas in (a), (b), and (d) indicate CCH (see Figure 4b). The dark black line in (a), (b), and (d) encloses the domain of the TP.



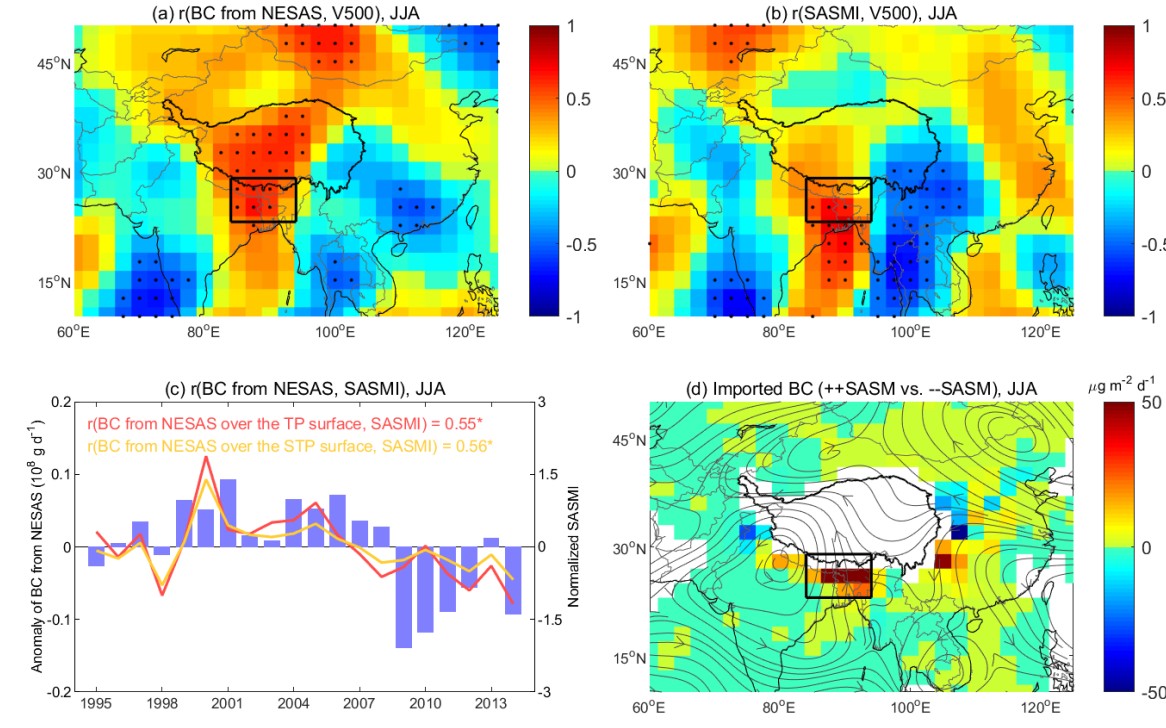

Figure 11. Connection between BC transport from South Asia to the TP surface and the SASM in summer. (a) Correlation coefficients (*r*) between meridional wind at 500 hPa (V500) and imported BC from northeastern South Asia (NESAS) over the TP surface. (b) *r* between the SASMI and V500. (c) Interannual variations in the intensity of the SASM, and imported BC from NESAS to the surface of the TP and southern TP (STP). (d) Differences in BC transport to the TP surface between the years with strong and weak SASM. Dots in (a) and (b) indicate the *r* in the corresponding grid is statistically significant ($p<0.05$). A *r* with '*' in (c) indicates that the *r* is statistically significant (p<0.05). Streamlines in (d) are the differences between the years with strong and weak SASM at 500 hPa. Boxed areas in (a), (b), and (d) indicate NESAS (see Figure 4b). The dark black line in (a), (b), and (d) encloses the domain of the TP.



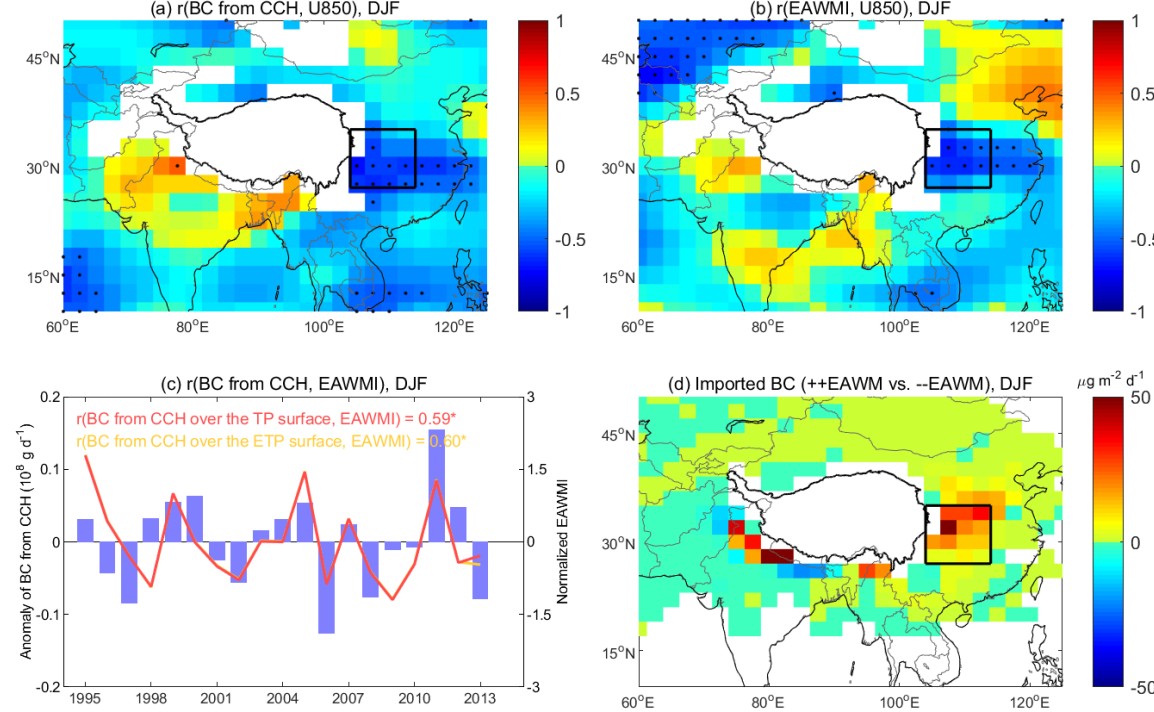

Figure 12. Connection between BC transport from East Asia to the TP surface and the EAWM in winter. (a) Correlation coefficients ($r$) between imported BC from central China (CCH) to surface BC in the TP and zonal wind at 850 hPa (U850). (b) $r$ between the EAWMI and U850. (c) Interannual variations in the intensity of the EAWM, and imported BC from CCH to the surface of the TP and eastern TP (ETP). (d) Differences in BC transport to the TP surface between the years with strong and weak EAWM. Dots in (a) and (b) indicate the $r$ in the corresponding grid is statistically significant ($p<0.05$). The unfilled grids in (a) and (b) are due to the topography. A $r$ with '*' in (c) indicates that the $r$ is statistically significant ($p<0.05$). Boxed areas in (a), (b), and (d) indicate CCH (see Figure 4d). The dark black line in (a), (b), and (d) encloses the domain of the TP.

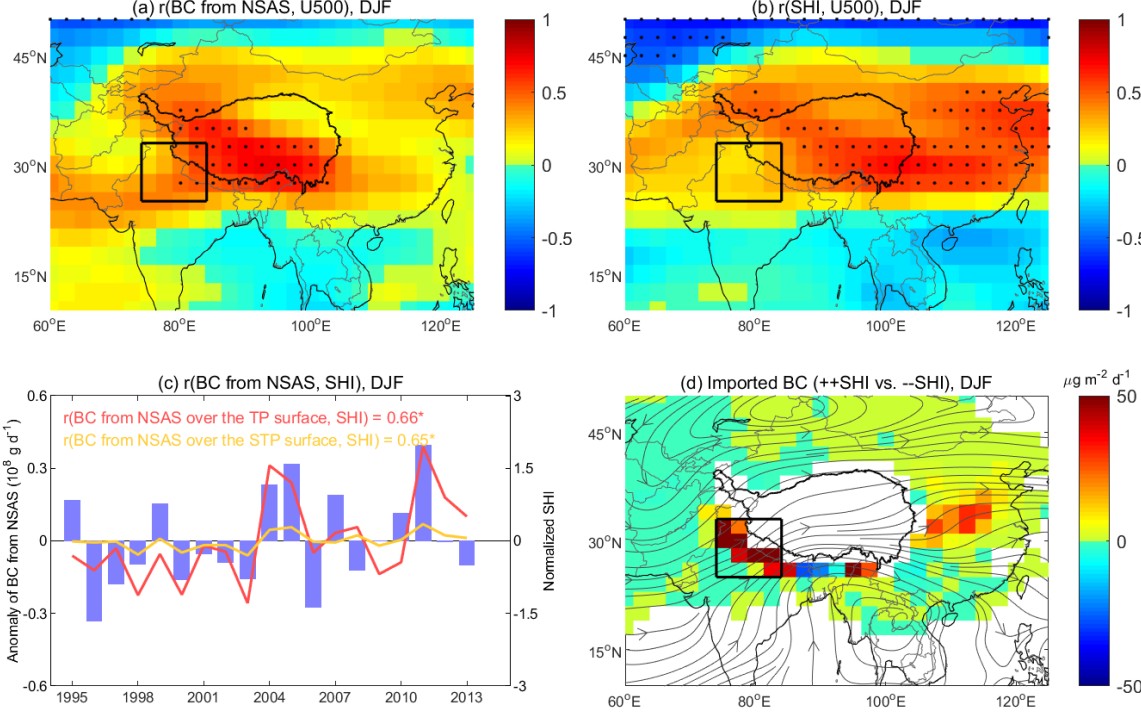

Figure 13. Connection between BC transport from South Asia to the TP surface and the Siberian High in
winter. (a) Correlation coefficients (*r*) between zonal wind at 500 hPa (U500) and imported BC from

northern South Asia (NSAS) over the TP surface. (b) *r* between the SHI and U500. (c) Interannual

variations in the SHI, and imported BC from NSAS to the surface of the TP and southern TP (STP). (d)

Differences in BC transport to the TP surface between the years with strong and weak Siberian High.

Dots in (a) and (b) indicate the *r* in the corresponding grid is statistically significant ($p<0.05$). A *r* with

'*' in (c) indicates that the *r* is statistically significant (p<0.05). Streamlines in (d) are the differences

between the years with strong and weak Siberian High at 500 hPa. Boxed areas in (a), (b), and (d)

indicate NSAS (see Figure 4d). The dark black line in (a), (b), and (d) encloses the domain of the TP.