# Peer review of "Impacts of atmospheric transport and biomass burning on the interannual variation in black carbon aerosols over the Tibetan Plateau"

_Atmospheric Chemistry and Physics, 2020_

## Referee Comment (RC1) · Anonymous Referee #2 · 1 Jun 2020

This study elucidated the impacts of meteorology and biomass burning on the seasonal and on the interannual variation in black carbon (BC) aerosols over the Tibetan Plateau (TP) based on 20-year GEOS-Chem simulations and HYSPLIT model. They found that over 90% surface BC in the TP comes from South Asia and East Asia. Both biomass burning and Asian monsoon played important roles in the variations in BC over TP. The results can contribute to the understanding of aerosols over TP and the manuscript is well written. However, there are still a few shortcomings that needs to be substantially revised.

General comments:

[Figure]

The induction section is vague without any quantitative description. From line 69 to 112, every sentence can be quantitatively expressed. Without these values from previous literatures, readers cannot fully get to background of this study.

There are some studies that explored the seasonal and interannual variation in BC over the TP, as the authors listed. The authors should compare the results in this study with those previous findings. Are they consistent? If not, why?

The authors only evaluated seasonal variation and spatial distribution of the BC. However, this study focused more on the interannual variation. Can the author find more data supporting the simulated interannual variability, like concentration or AAOD from surface measurement or satellite? Otherwise all the findings are based on model simulations and not fully convincible.

This study combined GEOS-Chem results and HYSPLIT model to attribution the BC from different source regions. Many studies did the sensitivity simulation with emissions from sources turned off. The emission perturbation method is more straightforward to me, since the backward trajectory method need to assume the decay time and the source region should be close to TP. What are the advantages and disadvantages between these two methods? In addition, I suggest the author to perform two more additional simulations with emissions from South Asia and East Asia turned off and compare the results with HYSPLIT outputs.

Specific comments:

The authors discussed a lot of the impacts from South Asia and East Asia. Southeast Asia has more biomass burning than South Asia. How the emissions from this region affect BC in TP?

Line 134: Anthropogenic emissions in 2000 were used. Since the past two decades, anthropogenic emissions have changed a lot, especially over South Asia and East Asia. The emissions out of date could cause large biases to the results. Also, the

biomass burning emissions only cover 1997–2011. The authors should discuss the uncertainties related to the emissions.

Line 136: The GEOS-Chem has a finer resolution over East Asia. Why the authors only use 2-degree version?

Line 178: It can also be due to the less emissions in 2000.

Line 193: The meteorological data for GEOS-Chem (MERRA) and HYSPLIT (NCEP/NCAR) are different. Will different meteorological data used in this study produce biases?

---

## Referee Comment (RC2) · Anonymous Referee #1 · 18 Jun 2020

This was an interesting work. The authors used GEOS-Chem model to investigate the origin of transported biomass burning aerosols over the Tibetan Plateau. Their results indicated the 47% of BC in the TP was from South Asia and 46% was from East Asia. Actually, I have suspected these quantitative results. Because the MODEL showed a quite coarse resolution of 2 X 2.5. So, this kind of evaluation probably had great uncertainties while the model in a such coarse grid space was used in a complex terrain (such as TP). A result based on the regional climate model evaluated the contribution of BC from South Asia was nearly 61% in monsoon season and 20% in non-monsoon season (Yang JGR, 2018). And another study that used a high resolution WRF-CHEM also indicated that finer resolution model could represent more reason-

able performance (Zhang ACP 2020, Impact of topography on black carbon transport to the southern Tibetan Plateau during the pre-monsoon season and its climatic implication). Their results found the complex topography in the model could generate 50% higher transport flux of BC in Himalayas. That's why I had such doubt need to explain by the authors. The manuscript was generally well written and comprehensive. I also have minor comments as follows. a. The definition for the source regions. Xinjiang in the northwestern China should be belong to CAS b. For the model performance evaluation, the comparison between in-site and model were not make sense. I did not believe the model output in a such great resolution could represent the surface situation. Probably a supplement if the authors want to show their model was credible.

---

## Author Comment (AC1) · 2 Sep 2020

*We thank the reviewers for their valuable comments and suggestions, which have helped us improve our manuscript in a great deal. We have made a revision accordingly. The point to point responses are provided below in Italic. The comparison of our manuscript between this version and the previous version is also provided.*

Anonymous Referee #2

This study elucidated the impacts of meteorology and biomass burning on the seasonal and on the interannual variation in black carbon (BC) aerosols over the Tibetan Plateau (TP) based on 20-year GEOS-Chem simulations and HYSPLIT model. They found that over 90% surface BC in the TP comes from South Asia and East Asia. Both biomass burning and Asian monsoon played important roles in the variations in BC over TP. The results can contribute to the understanding of aerosols over TP and the manuscript is well written. However, there are still a few shortcomings that needs to be substantially revised.

General comments:

The induction section is vague without any quantitative description. From line 69 to 112, every sentence can be quantitatively expressed. Without these values from previous literatures, readers cannot fully get to background of this study.

*Response: We have revised the introduction and made the introduction more specific and quantitative. Please see lines 57-121.*

There are some studies that explored the seasonal and interannual variation in BC over the TP, as the authors listed. The authors should compare the results in this study with those previous findings. Are they consistent? If not, why? The authors only evaluated seasonal variation and spatial distribution of the BC. However, this study

focused more on the interannual variation. Can the author find more data supporting the simulated interannual variability, like concentration or AAOD from surface measurement or satellite? Otherwise all the findings are based on model simulations and not fully convincible.

*Response: Thanks for all the points. Due to the harsh environment and sparse sites, BC observations in the TP are quite limited. Some observational studies investigated the seasonal variations in atmospheric BC over the TP (Marinoni et al., 2010; Cao et al., 2011; Zhao et al., 2012; Putero et al., 2014; Chen et al., 2018; Wang et al., 2018, 2019) and few have explored its interannual variation. Mao and Liao (2016) simulated the impacts of meteorology and emissions on the interannual variations of surface BC in the TP using a global chemical transport model. We evaluated the spatial and seasonal variations in BC concentrations over the TP from GEOS-Chem simulations against the observations obtained from previous studies (Table S1 and Figure S1). The comparison shows a good performance of the model, which was also suggested by previous studies (He et al., 2014b; K. Li et al., 2016).*

*Following the reviewer's suggestion, we further compared surface BC concentrations from GEOS-Chem simulations with those from MERRA2 reanalysis (The Modern-Era Retrospective analysis for Research and Applications version 2, M2TMNXAER, https://cmr.earthdata.nasa.gov/search/concepts/C1276812866-GES_DISC.html) (Figure S2 and Table S2). The magnitude and spatial distribution of BC concentrations in the TP from GEOS-Chem and MERRA2 are highly consistent, with correlation coefficients over 0.98 in the four seasons (Figure S2). BC concentrations from the two datasets show similar seasonal patterns over most areas in the TP (Table S2). The interannual variations of surface BC over the TP during 1995-2014 from the two datasets have strong similarity in spring, while weak similarity in winter (Table S2). Furthermore, surface BC from the two datasets shows similar seasonal and interannual patterns over East Asia, South Asia, and Southeast*

*Asia, except for the interannual variation in winter (Table S2). Please see the model evaluation in Supplement.*

*The estimate of the fractional contributions from various source regions to surface BC in the TP from this study is generally comparable with that from literature (Lu et al., 2012; Zhang et al., 2015; Yang et al., 2018). (1) The total contribution of South Asia and East Asia to surface BC in the TP is estimated to be 77%, between 69% by Zhang et al. (2015) and 84% by Lu et al. (2012). (2) The contribution of South Asia is stronger in winter and weaker in summer, which was also suggested by Lu et al. (2012), Zhang et al. (2015), and Yang et al. (2018). Yang et al. (2018) modeled that the contribution of South Asia is 61% in non-monsoon season (October-April) and 19% in monsoon season (May-September). (3) The estimate of local contribution is ~10%, comparable with that by Zhang et al. (2015), indicating the dominant role of BC transport from nonlocal regions. Nevertheless, there are some disagreements between this and previous studies in various extents. A noticeable disagreement is that this study estimates that the annual mean contribution of East Asia is approximately 35% (Figure 6, also see Figure S5), while the estimates by Lu et al. (2012) and Zhang et al. (2015) are respectively 17% and 19%. The discrepancy may be associated with the differences in region definitions and the estimation models. We added this information in this revision. Please see lines 497-510.*

This study combined GEOS-Chem results and HYSPLIT model to attribution the BC from different source regions. Many studies did the sensitivity simulation with emissions from sources turned off. The emission perturbation method is more straightforward to me, since the backward trajectory method need to assume the decay time and the source region should be close to TP. What are the advantages and disadvantages between these two methods? In addition, I suggest the author to perform two more additional simulations with emissions from South Asia and East Asia turned off and compare the results with HYSPLIT outputs.

*Response: Thanks for your suggestion. In this revision, we added the comparison between the backward-trajectory method developed in this study and the emission perturbation method (see Supplement). Estimations by the two methods are comparable (Figures 6 vs. S5). (1) The two methods both show that South Asia and East Asia are two dominant source regions of surface BC in the TP. The annual mean total contributions of South Asia and East Asia estimated by the two methods are close to each other, which are respectively 77% (Figure 6a) and 82% (Figure S5a). (2) The two methods modeled similar seasonal patterns of the contributions of South Asia and East Asia. The contribution of South Asia to surface BC in the TP is stronger in winter and spring, while the contribution of East Asia in stronger in summer and autumn. (3) The influences of BC transport on different TP subregions estimated by the two methods are generally consistent. For example, both methods show that over 70% of surface BC in the eastern TP comes from East Asia, and over 70% of surface BC in the southern TP comes from South Asia.*

*Overall, the backward-trajectory method developed in this study can reasonably quantify the relative contributions of different source regions to surface BC in the TP. This method has some other advantages. (1) The transport pathway of BC can be visibly expressed (Figures 2 and 3). (2) The spatial distribution of the contribution of source regions can be showed explicitly (Figures 3 and 4). (3) It is feasible for users of a chemical transport model to investigate the source-receptor relationships of BC if adjoint and tagged modes are unavailable to them. Yes, the reviewer is correct: this method assumes a fixed lifetime of atmospheric BC (i.e. 7 days in this study), which might lead to some uncertainties. The emission perturbation method does not require this assumption. It is reliable and straightforward. However, the emission perturbation method cannot provide the spatial information showed in Figures 2-4. It can only provide an overall assessment for the total contribution from each of the source regions defined, which are often rather limited, constrained by computational*

*cost.*

*We added these contents in the discussion section, please see lines 533-567.*

Specific comments:

The authors discussed a lot of the impacts from South Asia and East Asia. Southeast Asia has more biomass burning than South Asia. How the emissions from this region affect BC in TP?

*Response: Influenced by the prevailing westerlies, BC from Southeast Asia cannot be efficiently transported to the TP by atmospheric circulation. The annual mean contribution of Southeast Asia to surface BC in the TP is ~7% (Figure 6a). Seasonally, the contribution of Southeast Asia is highest in spring (10%) (Figure 6a) when it has the strongest fire BC emissions. Geographically, the contribution of Southeast Asia is highest in the southern TP (13% in the annual mean) (Figure 6c) because of its nearness to the TP. Please see our discussion in the revision in lines 344-353.*

Line 134: Anthropogenic emissions in 2000 were used. Since the past two decades, anthropogenic emissions have changed a lot, especially over South Asia and East Asia. The emissions out of date could cause large biases to the results. Also, the biomass burning emissions only cover 1997–2011. The authors should discuss the uncertainties related to the emissions.

*Response: Thanks for the comments. Figure S6 compares the anthropogenic emissions in 2000 used in this study and those in 2010 from the Task Force on Hemispheric Transport of Air Pollution Phase 2 (TF HTAP2). It shows that anthropogenic BC emissions in most areas of South Asia have substantially increased from 2000 to 2010 (Figure S6). On regional mean, anthropogenic BC emissions in East Asia have also increased from 2000 to 2010 (Figure S6). Inside East Asia, anthropogenic BC emissions in most regions of eastern China have increased, while decreased in Korea*

*and Japan from 2000 to 2010 (Figure S6). Therefore, our results may underestimate the BC transport from South Asia and East Asia to the TP in 2010. After 2010, anthropogenic BC emissions have changed as well (Zheng et al., 2018). Zheng et al. (2018) reported that anthropogenic BC emissions in China have reduced 27% from 2010 to 2017. BC emissions from biomass burning used in this study cover 1997-2011 and fire emissions in 2011 are used for simulation years after 2011. However, fire activities in South Asia have strong interannual variations and have been significantly increased from 2001 to 2016 (Earl et al., 2018). Therefore, fire emissions used here may lead to biases in the BC simulations after 2011. We have explained these uncertainties in Discussion. Please see lines 584-599.*

*Response: The evaluation of BC simulations has been moved to Supplement. The sentence has been revised there. Please see line 8 of the second paragraph in Supplement.*

Line 193: The meteorological data for GEOS-Chem (MERRA) and HYSPLIT (NCEP/NCAR) are different. Will different meteorological data used in this study produce biases?

*Response: Thanks for the points. In this study, GEOS-Chem BC simulations were driven by the MERRA data. The HYSPLIT backward trajectories were driven by the NCEP/NCAR meteorological fields. Using different input meteorological data in a numerical model may lead to some bias in the results. Nevertheless, we provide the following reasoning that the usage of different meteorological data in this study will not make systematical and substantial bias. (1) The wind field in MERRA and NCEP are comparable. Figures R1-R2 compared the wind fields in MERRA and NCEP data. The mean differences of zonal winds between the two datasets are around 0.6-2.4 m/s at 500 hPa and 0.6-1.2 m/s at 850 hPa over the TP and its surrounding regions (Figures R1-R2). The developed backward-trajectory method only considers the number of trajectories passing a grid. Therefore, minor errors in the exact location of*

*a trajectory within a grid of $2° \times 2.5°$ are tolerable. (2) The total number of the trajectories are large (4 times/day\*365 days\*20 years for 1 grid), so that the nonsystematic errors would be minimized after taking a mean of many trajectories. (3) The calculation of BC concentrations by GEOS-Chem and the calculation of backward trajectories by HYSPLIT are two independent processes and do not impact each other.*

[Figure]

Figure R1. Zonal winds at 500 hPa from NCEP (1st column) and MERRA (2nd column) meteorological data in 2005, and the difference between the two (3rd column). Values from MERRA data have been linearly interpolated to the NCEP grids.

[Figure]

Figure R2. Zonal winds at 850 hPa from NCEP (1st column) and MERRA (2nd column) meteorological data in 2005, and the difference between the two (3rd column). Values from MERRA data have been linearly interpolated to the NCEP grids.

[revised manuscript text omitted]

---

## Author Comment (AC2) · 2 Sep 2020

*We thank the reviewers for their valuable comments and suggestions, which have helped us improve our manuscript in a great deal. We have made a revision accordingly. The point to point responses are provided below in Italic. The comparison of our manuscript between this version and the previous version is also provided.*

Anonymous Referee #1

This was an interesting work. The authors used GEOS-Chem model to investigate the origin of transported biomass burning aerosols over the Tibetan Plateau. Their results indicated the 47% of BC in the TP was from South Asia and 46% was from East Asia. Actually, I have suspected these quantitative results. Because the MODEL showed a quite coarse resolution of 2 X 2.5. So, this kind of evaluation probably had great uncertainties while the model in a such coarse grid space was used in a complex terrain (such as TP). A result based on the regional climate model evaluated the contribution of BC from South Asia was nearly 61% in monsoon season and 20% in non-monsoon season (Yang JGR, 2018). And another study that used a high resolution WRF-CHEM also indicated that finer resolution model could represent more reasonable performance (Zhang ACP 2020, Impact of topography on black carbon transport to the southern Tibetan Plateau during the pre-monsoon season and its climatic implication). Their results found the complex topography in the model could generate 50% higher transport flux of BC in Himalayas. That's why I had such doubt need to explain by the authors. The manuscript was generally well written and comprehensive.

*Response: Thanks for your comments. Although GEOS-Chem's resolution is relatively coarse, the comparison with observations from previous studies suggests that GEOS-Chem can generally capture the spatial and seasonal variations of surface BC in the TP and its surrounding regions (Table S1 and Figure S1). We compared the seasonal*

*variations in BC simulations with those in literature over different TP subregions. The comparison shows reasonable agreement between the simulations and observations. Please see the comparison and discussions in the supplement and in lines 170-180.*

*In this revision, we further ran a set of simulations using the emission perturbation method (see Supplement). The comparisons between the emission perturbation method and the backward-trajectory method developed in this study are reasonably well, suggesting the estimates from the backward-trajectory method can reasonably quantify the relative contributions of different source regions to surface BC in the TP.*

*We also compared the results from this study with literature (Lu et al., 2012; Zhang et al., 2015; Yang et al., 2018) and we found a general agreement between the two. (1) We estimated that the annual mean total contribution of South Asia and East Asia is 77%, and the estimates by Lu et al. (2012) and Zhang et al. (2015) are respectively 84% and 69%. (2) Our results showed that the contribution of South Asia is stronger in winter and weaker in summer, which was also suggested by Lu et al. (2012), Zhang et al. (2015), and Yang et al. (2018). Yang et al. (2018) modeled that the contribution of South Asia is 61% in non-monsoon season (October-April) and 19% in monsoon season (May-September). (3) The estimate of local contribution in this study and in Zhang et al. (2015) is both around 10%, indicating the dominant role of BC transport from nonlocal regions regarding the origins of the BC in the TP. Our results disagree with some of previous studies in various details. For instance, we estimated that the annual mean contribution of East Asia is approximately 35%, while the estimates by Lu et al. (2012) and Zhang et al. (2015) are respectively 17% and 19%. The discrepancy may be associated with the differences in region definitions and the estimation models. Please see discussions in lines 497-510 and 533-567.*

*In this study, the simulations are with a horizontal resolution of 2° latitude by 2.5° longitude. Such a resolution may not fully capture processes in the sub-grid scale, such as the mountain-valley wind (Cong et al., 2015). Using a regional model, Zhang et al. (2020) demonstrated that compared with simulations with a coarser resolution, simulations with higher resolution can better resolve the effects of topography and consequently get stronger transport flux of BC from South Asia to the TP. Therefore, our results may somewhat underestimate the contribution of South Asia because of the model resolution. Using regional models at higher resolutions in the future can better describe the terrain effect in the TP. Please see the discussions in lines 575-582.*

I also have minor comments as follows.

a. The definition for the source regions. Xinjiang in the northwestern China should be belong to CAS

*Response: Thanks for your suggestion. In this revision, Xinjiang and part of Mongolia are defined as areas in Central Asia (see Figure 1b).*

b. For the model performance evaluation, the comparison between in-site and model were not make sense. I did not believe the model output in a such great resolution could represent the surface situation. Probably a supplement if the authors want to show their model was credible.

*Response: Thanks for the points. We have moved the comparison to Supplement. It is true that an observation can be compared with the model result in a grid only if the condition in that grid is homogenous. This may be somewhat the case at some remote and rural sites in the TP. The performance of GEOS-Chem in simulating BC over the TP and worldwide has been evaluated by scientists using in situ measurements of BC in surface air, and BC absorption aerosol optical depth (Kopacz et al., 2011; He et al., 2014, Li et al.,2015). He et al. (2014) found that the simulated BC in surface air are compared statistically well with observations at sites away from urban areas and the model can generally capture the seasonality of the observations, whereas the BC concentrations in the TP are likely to be underestimated by the model. We further evaluate the spatial and seasonal variations in GEOS-Chem BC simulations using observational data from literature and found a significant correlation (r = 0.99,*

*p<0.05) between the observations and simulations at the rural and remote sites (Figure S1).*

*In this revision, we further compared surface BC concentrations, in terms of seasonal and interannual variations, from the GEOS-Chem simulations with those from MERRA2 reanalysis (The Modern-Era Retrospective analysis for Research and Applications version 2, M2TMNXAER, https://cmr.earthdata.nasa.gov/search/concepts/C1276812866-GES_DISC.html). The magnitude and spatial distribution of surface BC concentrations in the TP from GEOS-Chem and MERRA2 are highly consistent, with correlation coefficients over 0.98 in the four seasons (Figure S2). Furthermore, surface BC from GEOS-Chem and MERRA2 shows similar seasonal patterns over the TP as well as over East Asia, South Asia, and Southeast Asia (Table S2). For the interannual variations of surface BC over these regions, the similarity between the simulations and reanalysis is strong in spring, summer, and autumn, while is weak in winter (Table S2). Please see the discussions in Supplement and lines 170-189.*

*References*

[revised manuscript text omitted]

---

## Author Response (AR2)

*Dear Prof. Jianzhong Ma,*

*Thank you for your valuable comments and suggestions. We have made a revision accordingly. The point to point responses are provided below in Italic. The*

*comparison of our manuscript between this version and the previous version is also provided. Please let us know should you have future questions.*

*Thanks again for your time and efforts on our paper.*

*Best regards,*
*Jane Liu on behalf of all the co-authors*

Comments to the Author:

Dear Prof. Jane Liu and all authors,

I had sent the revised manuscript to two referees for further review, but one of them did not give his/her review report until now. I think your manuscript can be accepted after the following issues have been addressed.

Figure 2b shows the 7-day backward trajectories (in meters above the ground) arriving at the TP surface. Since no trajectories are shown for the domain within the TP, it is difficult to see the transport pathways of air masses arriving at a specific region, e.g., the central TP. Air masses from the west (blue dots) are still at high altitudes (above 1000 m) at the western edge of the TP. Are 1-day backward trajectories for these air masses included in the plot? Will these air masses enter into the TP domain from the west or other side(s) of the TP? The authors may consider revising Fig. 2b and/or providing additional discussion for better understanding this subpanel.

While Fig. 2a presents low surface BC concentrations on the north of the TP, Fig. 2c shows high BC transported fluxes there. How about the regional distribution of BC column density? Or, does an elevated BC layer existing on the north of the TP?

Best regards, Jianzhong

*Response:*

*Figure 2 is shown to be an example of the estimate of BC transport to the TP surface and to help to understand the calculation of BC transport flux to the TP*
*surface and its spatial distribution.*

*Following the editor's advice, we replaced the Figure 2a in the last version with a new figure showing BC total column, and replaced Figure 2b with a new figure showing all 7-day backward trajectories arriving at the TP surface in the month. The distribution of surface BC concentrations is moved to Supplement.*

[revised manuscript text omitted]